# Using subthreshold events to characterize the functional architecture of the electrically coupled inferior olive network

Yaara Lefler[1,2†‡*], Oren Amsalem[1,2†*], Nora Vrieler[1,2], Idan Segev[1,2], Yosef Yarom[1,2]

[1]Department of Neurobiology, Institute of Life Sciences, The Hebrew University of Jerusalem, Jerusalem, Israel; [2]Edmond and Lily Safra Center for Brain Sciences, The Hebrew University of Jerusalem, Jerusalem, Israel

**Abstract** The electrical connectivity in the inferior olive (IO) nucleus plays an important role in generating well-timed spiking activity. Here we combined electrophysiological and computational approaches to assess the functional organization of the IO nucleus in mice. Spontaneous fast and slow subthreshold events were commonly encountered during in vitro recordings. We show that whereas the fast events represent intrinsic regenerative activity, the slow events reflect the electrical connectivity between neurons ('spikelets'). Recordings from cell pairs revealed the synchronized occurrence of distinct groups of spikelets; their rate and distribution enabled an accurate estimation of the number of connected cells and is suggestive of a clustered organization. This study thus provides a new perspective on the functional and structural organization of the olivary nucleus and a novel experimental and theoretical approach to study electrically coupled networks.

**\*For correspondence:**
yaara.lefler@mail.huji.ac.il (YL);
oren.amsalem1@mail.huji.ac.il (OA)

†These authors contributed equally to this work

Present address: ‡UCL Sainsbury Wellcome Centre for Neural Circuits and Behaviour, London, United Kingdom

**Competing interests:** The authors declare that no competing interests exist.

## Introduction

In recent years, research has confirmed that electrically coupled neural networks are found in every major region of the central nervous system (*Condorelli et al., 2000*; *Bennett and Zukin, 2004*; *Connors and Long, 2004*; *Hormuzdi et al., 2004*). One common feature of these networks is their synchronized rhythmic activity (*Connors and Long, 2004*; *Bennett and Zukin, 2004*; *Connors, 2017*; *Coulon and Landisman, 2017*), which has been shown to be correlated with higher brain functions such as states of arousal, awareness, cognition and attention (*Ritz and Sejnowski, 1997*; *Engel et al., 2001*; *Buzsáki, 2005*; *Steriade, 2006*; *Uhlhaas et al., 2009*; *Wang, 2010*). Recently, it has been demonstrated that the efficiency of electrical synapses is modulated by electrical and chemical activity, very much like chemical synapses (*O'Brien, 2014*; *Marder et al., 2017*; *Coulon and Landisman, 2017*). It thus stands to reason that the functional architecture of these networks must undergo continuous modification to meet the system's demands. This underscores the urgent need to determine the functional state of a network and associate it with the corresponding brain states. Since anatomical information is insufficient, this can only be done using physiological parameters that capture the functional architecture of a network at any given time.

The inferior olive network, which was among the first electrically coupled networks to be studied in the mammalian brain, provides primary excitatory input to the cerebellar cortex (*Eccles et al., 1966*). There is a general consensus that the function of this network is to generate synchronous activity in olivary neurons, which provide temporal information for either learning processes, motor execution, sensory predictions or expectations (*Llinás and Sasaki, 1989*; *Lou and Bloedel, 1992*; *Welsh et al., 1995*; *Van Der Giessen et al., 2008*; *Llinás, 2009*; *De Zeeuw et al., 2011*; *Ohmae and Medina, 2015*; *Heffley et al., 2018*). Temporal information is thought to be generated

by the subthreshold sinusoidal-like oscillations of the membrane voltage that appear to emerge from an interplay between the membrane properties and network connectivity (*Llinás and Yarom, 1986*; *Lampl and Yarom, 1997*; *Manor et al., 1997*; *Loewenstein et al., 2001*; *Devor and Yarom, 2002b*). Recently, this oscillatory activity was shown to be governed by chemical synaptic inputs that partially originate in the deep cerebellar nuclei and modulate the efficacy of the coupling by defining the spatial extent of the electrically coupled network (*Lefler et al., 2014*; *Mathy et al., 2014*; *Turecek et al., 2014*).

Early work on the morphological organization of the IO indicated that it is organized in clusters of up to eight neurons, whose dendrites are integrated in glomerulus structures (*Sotelo et al., 1974*) and are innervated by both excitatory and inhibitory synaptic inputs (*de Zeeuw et al., 1990*; *de Zeeuw et al., 1989*). This presumed clustered organization has been supported by dye coupling studies showing that each olivary neuron is anatomically coupled to roughly ten other neurons (*Devor and Yarom, 2002a*; *Leznik and Llinás, 2005*; *Placantonakis et al., 2006*; *Hoge et al., 2011*; *Turecek et al., 2014*), and by a recent detailed morphological study demonstrating the directionality of IO neuron dendrites (*Vrieler et al., 2019*). Physiologically however, the organization of the network has only been addressed in a few voltage-sensitive dye imaging studies which found ensembles of synchronously active neurons corresponding to a cluster size estimation of hundreds of neurons (*Devor and Yarom, 2002b*; *Leznik et al., 2002*). The documented synchronicity of complex spike activity in tens to hundreds of cerebellar Purkinje cells during motor tasks and sensory stimulation is also in favor of such ensemble organization (*Bloedel and Ebner, 1984*; *Welsh et al., 1995*; *Mukamel et al., 2009*; *Ozden et al., 2009*; *Schultz et al., 2009*; *De Zeeuw et al., 2011*; *Byk et al., 2019*; *Kostadinov et al., 2019*).

In this study, we describe a novel method to estimate the size and connectivity of a network by analyzing the all-or-none subthreshold unitary events known as 'spikelet'. Initially, spikelets were considered as the manifestation of an action potential transmitted via electrical synapses (*Llinas et al., 1974*; *MacVicar and Dudek, 1981*; *Valiante et al., 1995*; *Galarreta and Hestrin, 1999*; *Gibson et al., 1999*; *Mann-Metzer and Yarom, 1999*; *Hughes et al., 2002*; *Chorev and Brecht, 2012*). However other studies have also referred to spikelets as reflecting either local dendritic regenerative responses (*Spencer and Kandel, 1961*; *Golding and Spruston, 1998*; *Smith et al., 2013*); action potentials in the initial segment or at an ectopic site along the axon that fail to invade the soma (*Stasheff et al., 1993*; *Avoli et al., 1998*; *Juszczak and Swiergiel, 2009*; *Sheffield et al., 2011*; *Dugladze et al., 2012*; *Michalikova et al., 2017*); electrical coupling between axons (*Schmitz et al., 2001*; *Traub et al., 2002*); or extracellularly recorded activity of nearby neurons (*Vigmond et al., 1997*; *Scholl et al., 2015*). Here we show that the spontaneous unitary events recorded from olivary neurons can be classified into two groups that differ in their waveform and properties: fast events having identical waveforms with variable high amplitudes, and slow events having different waveforms and low amplitudes. We show that the low-amplitude slow events reflect the occurrence of action potentials in electrically coupled neurons, whereas the high-amplitude fast events are likely to represent internal regenerative responses. We then used slow events recorded simultaneously in pairs of neurons to estimate the size of the network (i.e. the number of neurons that are connected to each neuron) and the network connectivity profile. We found that each olivary neuron is electrically connected to an average of 19 other neurons and that the network is not randomly connected but rather composed of functional clusters of connected neurons.

## Results

### Spontaneous unitary events recorded in neurons of the inferior olive

The subthreshold spontaneous activity recorded from IO neurons (*Figure 1A*) is composed of unitary unipolar events of varying amplitudes and waveforms. Such events were observed in 74.3% of the neurons (188 out of 253) with an average rate of $0.7 \pm 0.6$ Hz (calculated in 70 neurons). The subthreshold events could readily be divided into two populations of small and large events (*Figure 1A* inset, circles vs. stars), as shown by the amplitude histogram (*Figure 1B*). In this example neuron, K-means clustering of the events' waveforms reveals five distinct groups (*Figure 1B,C*), which when normalized (*Figure 1D*), showed the waveform difference between the two types; one type had high amplitude and fast kinetics, and the second type had low amplitude and slow kinetics.

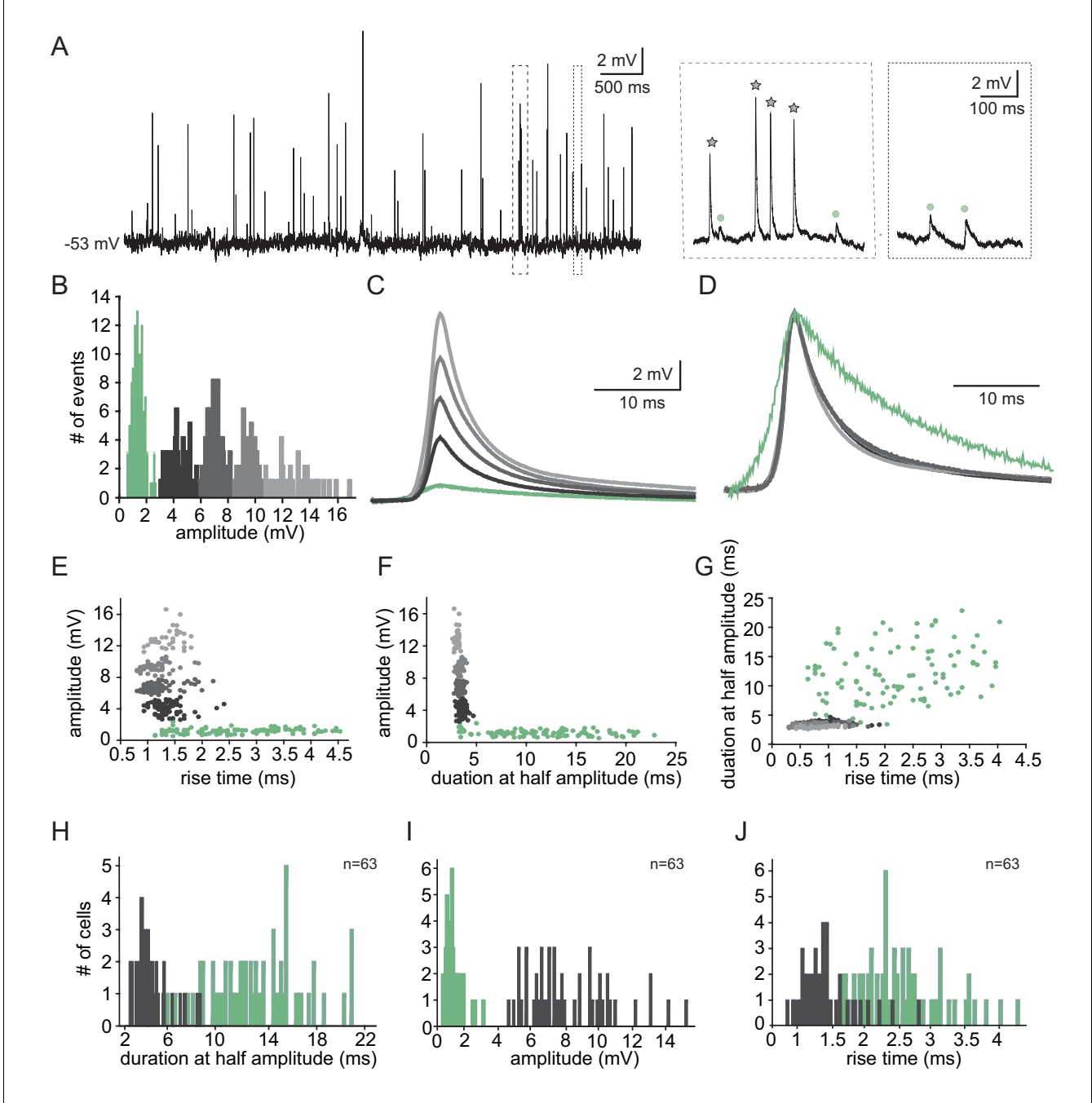

**Figure 1.** Two types of subthreshold spontaneous events recorded in olivary neurons. (**A**) Spontaneous subthreshold events recorded from an olivary neuron. Right panels - higher magnification of the marked rectangles; gray stars - fast and high amplitude events; green circles - slow and small events. (**B**) The distribution of the events' amplitudes in this neuron; colors were assigned according to the K-means analysis of the amplitudes. (**C**) Averages of the subthreshold events in each cluster, color coded as in B. (**D**) The normalized events shown in C. (**E–G**) Scatter plots for the relationships between the shape indices of the subthreshold events (color coded as in B). (**E**) Amplitude and rise time; (**F**) Amplitude and half width; (**G**) half width and rise time. (**H–J**) Histograms of the shape indices (half width (**H**); amplitude (**I**); and rise time (**J**)) of the subthreshold events in a population of 63 olivary neurons; green and gray bars correspond to slow and fast events respectively.

The online version of this article includes the following source data for figure 1:

**Source data 1.** Source data for *Figure 1*.
**Source data 2.** Trace for *Figure 1A*.

To further analyze the event waveforms, we measured each event's rise time and duration at half amplitude. The results obtained from a representative neuron are summarized in *Figure 1E–G*. One type (black to gray circles) had a relatively high amplitude (2.4–16.6 mV; average of 7.5 ± 3.1 mV) and fast kinetics (average rise time of 1.3 ± 0.3 ms and average half duration of 3.4 ± 0.3 ms) whereas the second type (green circles) had a relatively low amplitude (0.6–1.9 mV; average of 1.17 ± 0.3 mV) and slow kinetics (average rise time of 2.4 ± 0.4 ms and average half duration of 11.8 ± 5 ms). Plotting the duration as a function of the rise time (*Figure 1G*), which further supports the two-type scheme, failed to demonstrate a monotonous relationship between the rise-time and half-width that is expected from different dendritic locations of synapses (Rall's cable theory; *Rall, 1967*). Thus, it seems unlikely that the two types represent signals arising from different locations along the cell's morphological structure. The distribution of rise-time and half-width in a population of 63 neurons (of which 49 neurons had the two types of events), which is summarized in *Figure 1H–J*, confirms that there were indeed two distinct types of events. Whereas the high-amplitude events had a fast rise time (0.8–2.8 ms; average of 1.4 ± 0.4 ms) and short duration (2.5–8.3 ms; average of 4.2 ± 1.3 ms), the low-amplitude events had a longer rise time (1.3–4.3 ms; average of 2.5 ± 0.6 ms) and a longer duration (3.6–21 ms; average of 12.7 ± 3.9 ms). For the high-amplitude events, the broad distribution of amplitudes (ranging from 4.5 to 15.3 mV) and the somewhat limited distribution of rise-times and durations strongly indicates that these groups of fast events were generated by a similar mechanism. Overall, the frequency of slow events was four times higher (0.56 ± 0.62 Hz; n = 69 neurons) than that of the fast events (0.14 ± 0.18 Hz; n = 58 neurons).

To further distinguish between these two events, we examined the effect of the membrane voltage on the occurrence and waveforms of both types of unitary events. To that end, we used DC current injection, which on average set the membrane potential to a range of −33 to −90 mV. *Figure 2A–B* shows the aligned superimposed traces of slow events (*Figure 2A*) and fast events (*Figure 2B*) from one neuron. Normalizing the event amplitudes (*Figure 2A–B*, right panels) shows that whereas the shape of the slow events was unaffected by the current injection (A), the fast events showed a slowdown of the late repolarizing phase with hyperpolarization (B). Quantifying the effect of the injected current (see Materials and methods) on the amplitude and duration at 20% of the amplitude in 16 neurons revealed no effect on the amplitude of either type of events (*Figure 2C,D*; $R^2$ = 0.0057 and 0.0035, respectively). The duration of the slow events was slightly, but not significantly, affected (*Figure 2E*; $R^2$ = 0.44; one-sample t-test p=0.057). In contrast, DC current injection significantly increased the duration of the fast events (*Figure 2F*; $R^2$ = 0.712; one-sample t-test p=0.0005). Comparing the two sets of data revealed a significant difference (*Figure 2E* vs. *Figure 2F*; paired t-test p=0.017). Finally, we measured the effect of the DC current injection on the rate of occurrence of the subthreshold unitary events (*Figure 2G–H*). Whereas the frequency of the slow events remained unaffected (*Figure 3G*, $R^2$ = 0.012), the frequency of the fast events increased by a factor of up to 5 (*Figure 2H*; $R^2$ = 0.836). This difference between the occurrence of slow and the fast events, which was highly significant (*Figure 2G* vs. *Figure 2H*, paired t-test p=0.005), further supports our presumption that two different mechanisms generate the two types of unitary subthreshold events. Since the membrane potential did not change the amplitude of the events, it implies that neither of them represents chemical synaptic potentials.

However, application of excitatory synaptic blockers (DNQX and APV) completely eliminated the presence of the fast events (frequency before application was 0.056 ± 0.066 Hz, with zero events after application in n = 5 cells). Thus, it is likely that the fast events reflect intrinsic regenerative response that is triggered by excitatory synaptic input (as for the effect on slow events, see below).

## Slow events reflect electrical coupling between neurons

Whole-cell recordings from pairs of coupled olivary neurons revealed that the post-junctional responses to both spontaneous (*Figure 3A*) and evoked (*Figure 3B*) action potentials in one neuron were precisely correlated with depolarizing events in the coupled neuron. Both the spontaneous and the evoked events resembled the spontaneously recorded slow events depicted in *Figure 1*. These events had an amplitude of 1.2 ± 0.12 mV, a rise time of 3.7 ± 0.9 ms and a duration of 14.9 ± 4.0 ms, thus well within the range of spontaneously measured slow events. Paired recordings from 30 neurons showed that the amplitudes of the events varied from 0.12 to 1.40 mV (average of 0.61 ± 0.35 mV) whereas the average rise times and half durations were 2.61 ± 1.09 ms and 14.31 ± 6.72 ms, respectively, in line with the measured distribution of spontaneously occurring slow

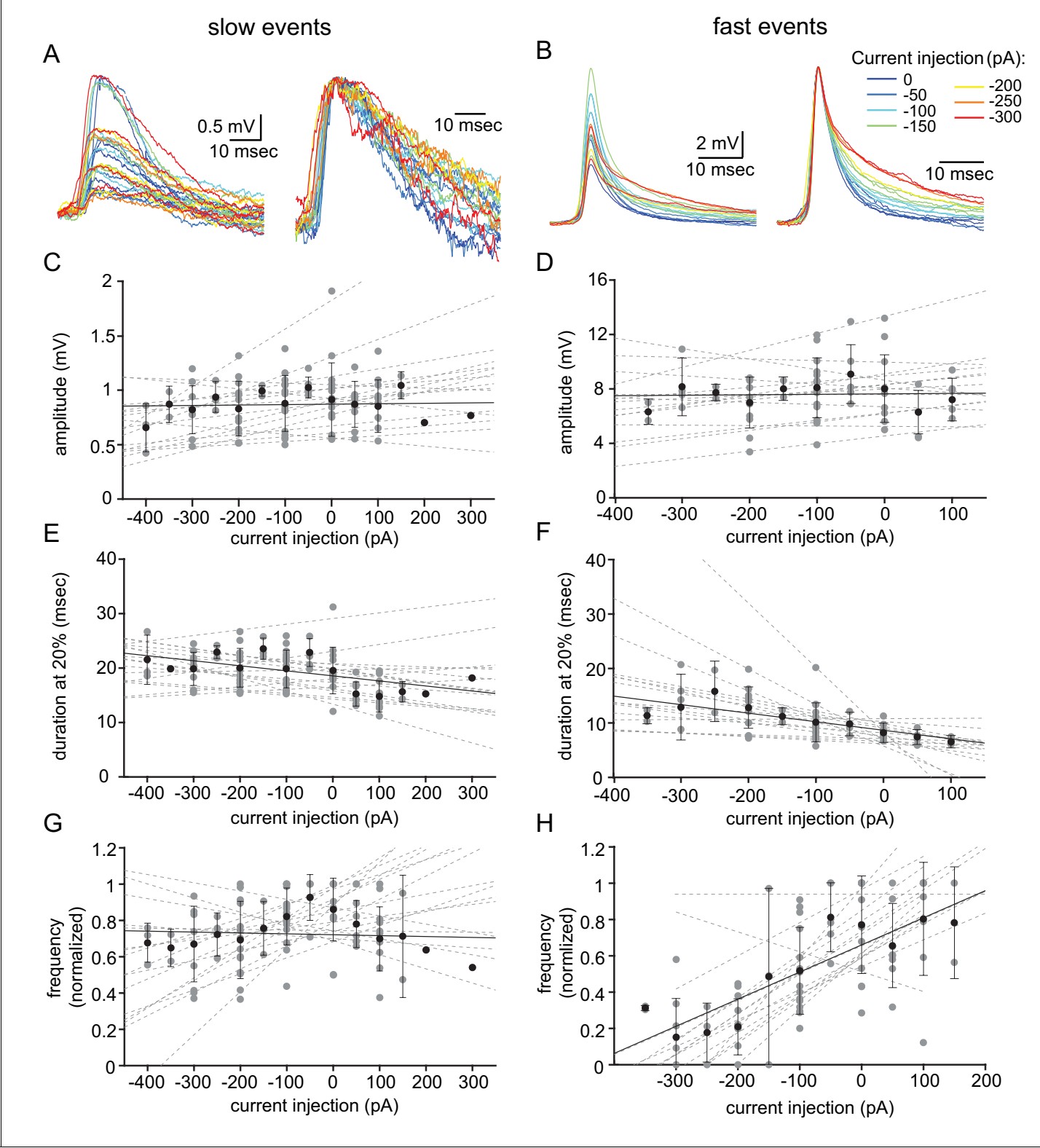

**Figure 2.** Voltage dependency of the subthreshold events. (**A**) Superimposed slow events recorded during seven different DC current injections (−300 to 0 pA, color coded, left panel), and normalized by amplitude (right panel). (**B**) Same as A, for the fast events. (**C–H**) The effect of DC current injection on the amplitude (**C–D**), the duration measured at 20% of the amplitude (**E–F**) and the frequency of occurrence (**G–H**) measured in 16 neurons. Gray circles represent the average data from individual neurons, each fitted with a linear regression (dashed gray lines). Black circles and error bars (std)

*Figure 2 continued on next page*

*Figure 2 continued*

represent the average value for all the neurons in each current injection. Note that the decrease in duration (F) and the increase in frequency (H) with depolarization only occurs for fast events.

The online version of this article includes the following source data for figure 2:

**Source data 1.** Source data for *Figure 2*.

events (*Figure 3—figure supplement 1C*, for rise time and duration p=0.58 and p=0.15 respectively, paired t-test).

In order to identify the source of the spontaneously occurring slow events, we characterized and compared three types of subthreshold events: the spontaneous slow events; the events triggered by action potentials in pair recording and the chemical synaptic potentials triggered by ChR activation in Thy1 mice. The responses to minimal light intensity is shown in *Figure 3—figure supplement 1B*. The shapes of the evoked synaptic events differed significantly from those of the spontaneous slow events in the same neurons (p<0.002 for all comparisons (rise time, half duration and amplitudes), paired t-test, n = 17 cells), as well as from evoked slow events measured in pair recordings (*Figure 3—figure supplement 1C*, black circles; p<0.002 for all comparisons, paired t-test). This strongly suggests that the small and slow spontaneous events represent action potentials occurring in electrically coupled neurons. To further support this possibility, we measured the effect of excitatory synaptic blockers on the occurrence of the slow events. The chemical synaptic blockers drastically reduced the frequency of the slow events (from $2.27 \pm 1.75$ Hz to $0.068 \pm 0.078$ Hz) while amplitude, rise time and duration at half amplitude were not affected (p=0.8, 0.25 and 0.96 respectively; paired KS test; *Figure 3—figure supplement 1E*). In addition, evoked slow events could readily be seen in pair recordings with DNQX (*Figure 3—figure supplement 1D*, n = 10 cells). The decrease in the frequency of the slow events could suggest that a subpopulation of slow events represent chemical synaptic potentials. However, the spontaneous spiking activity of olivary neurons is triggered by the fast events (*Figure 3—figure supplement 2*), that are completely eliminated in the presence of chemical synaptic blockers (see above), causing a drastic reduction in spiking activity and their electrical posts-junctional presentation, the slow events.

We conclude that the slow events represent the post-junctional responses to action potentials in coupled cells and therefore we refer to them as 'spikelets'.

The relatively broad range of spikelet parameters (*Figure 1*) can be attributed to a wide range of coupling strengths, different locations of the gap junctions along the dendritic structure or different durations and shapes of the pre-junctional action potential, which is a well-known feature of olivary action potentials (*Llinás and Yarom, 1981a*; *Llinás and Yarom, 1981b*). We first examined the effect of coupling strength by calculating the ratio of the amplitudes of the pre-junctional action potential to the post-junctional spikelet and compared it to the coupling coefficient measured by direct current injection (see Materials and methods). As shown in *Figure 3C*, there was a significant positive correlation (with a slope of 0.134; $R^2 = 0.614$, p<0.0001; Pearson correlation). Next, we examined the effect of the shape of the pre-junctional action potential on the spikelet parameters. To that end, we partially blocked the voltage dependent potassium current by adding TEA (10 mM) to the bath solution. In the presence of TEA, a variety of action potential waveforms were elicited by current injection (*Figure 3D*). In particular, the initial upstroke of the action potential was unaffected, but there was a significant broadening of the repolarizing phase (*Figure 3D*, upper panel, blue) that often elicited a second calcium-dependent action potential (*Figure 3D*, upper panel, cyan). This variety of action potential waveforms was always associated with electrical post-junctional responses that could be clustered into two distinct groups (*Figure 3D*, lower panel). The prolongation of the action potential was, as expected, followed by a matching increase in the duration of the post-junctional responses (*Figure 3D*, lower panel, blue traces). The appearance of the second component was associated with a slow wave of depolarization in the post-junctional cell (cyan traces). This suggests that the wide range of spikelet parameters (*Figure 1*) can be accounted for by the variability in coupling strength and pre-junctional action potential waveforms.

Finally, we examined the occurrence of spikelets in neurons that exhibited subthreshold oscillatory activity. Since these oscillations occurred simultaneously in several neurons (*Lefler et al., 2013*), it was expected that spikelet occurrence will be correlated with the oscillatory activity. About

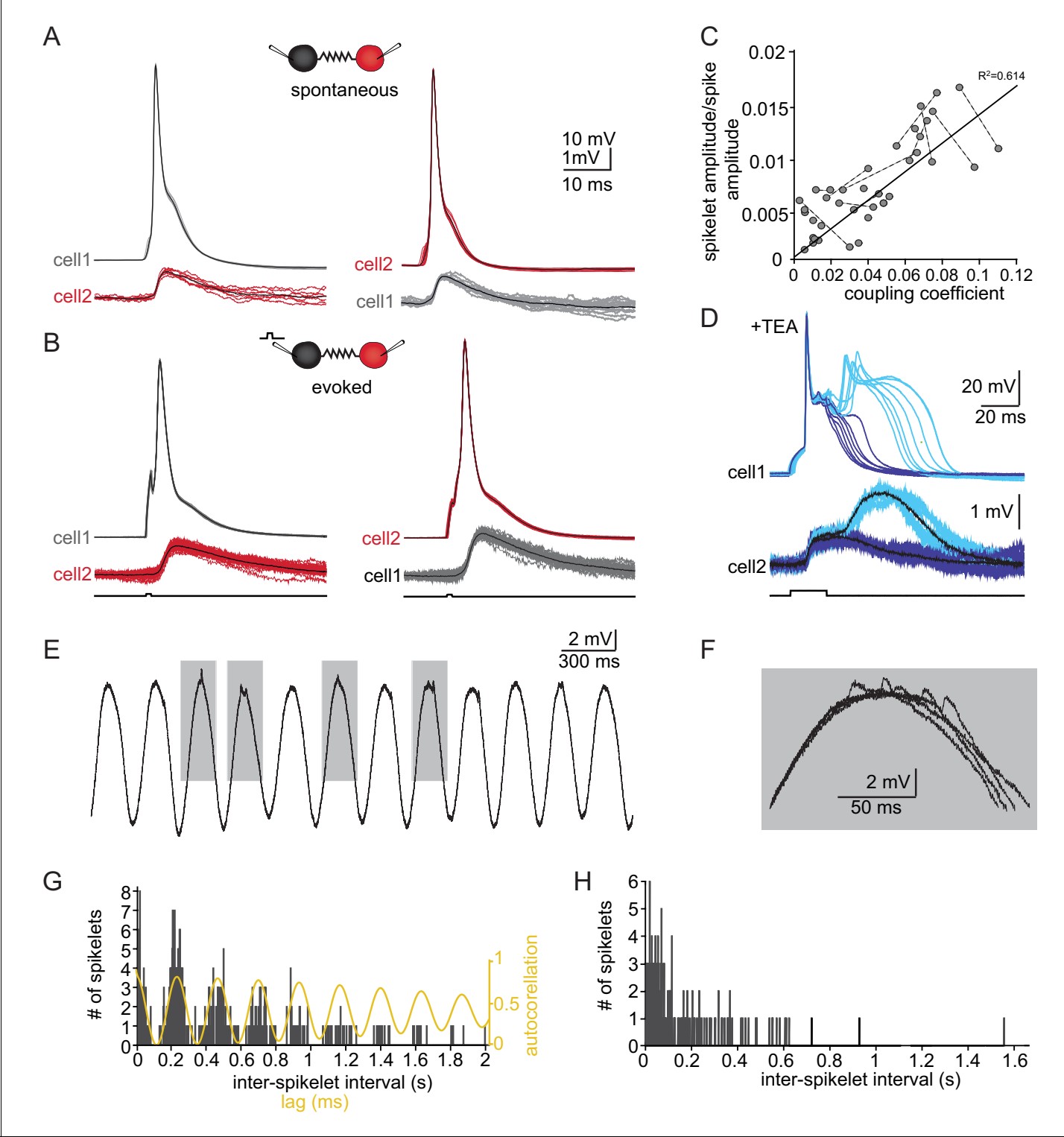

**Figure 3.** The slow events represent the electrical coupling between neurons. (**A**) Superimposed traces of spontaneously occurring action potentials recorded simultaneously from a pair of coupled neurons (red and gray traces; black traces represent the average events). (**B**) The same as in A for action potentials evoked by 100 pA, 1 ms current pulses. (**C**) The linear relationship between the DC coupling coefficient and the spike coupling coefficient. Pairs of cells are connected by dashed lines. Blue line is the linear regression fit ($R^2$ = 0.6). (**D**) Paired recording in the presence of 10 mM TEA. Action potentials with relatively long durations (upper panel, blue traces) were elicited in cell 1 by 50 pA, 20 ms current pulse. Occasionally they were followed by a second response (cyan traces). These action potentials elicited post junctional responses in cell 2 with corresponding waveforms

*Figure 3 continued on next page*

*Figure 3 continued*

(lower traces). (**E**) Subthreshold events recorded in oscillating olivary neuron. (**F**) Superposition of the gray rectangles in E, at higher magnification. Note that spikelets were only present for 50 ms along the peak of the oscillations. (**G**) Inter-spikelet interval (ISLI) from the same neuron (using 4 ms bins), and an autocorrelation (yellow line) of the membrane potential. (**H**) The ISLI distribution in a non-oscillating neuron.

The online version of this article includes the following source data and figure supplement(s) for figure 3:

**Source data 1.** Source data for *Figure 3*.
**Source data 2.** Trace for *Figure 3A*.
**Source data 3.** Trace for *Figure 3B*.
**Source data 4.** Trace for *Figure 3E*.
**Figure supplement 1.** The slow events represent electrical coupling between olivary neurons.
**Figure supplement 2.** The fast evens are the major source of spontaneous action potentials in olivary neurons.

50% of the olivary neurons showed spontaneous subthreshold oscillations (*Figure 3E*). Careful examination of the peaks of the oscillations (*Figure 3F*) revealed that they were crowned with spikelets. To quantify this observation, we calculated the distribution of the inter-spikelet-interval (ISLI, *Figure 3G*, black bars), and found distinct groups appearing at intervals of 200 ms. We then calculated the autocorrelation function of the subthreshold oscillations (*Figure 3G*, yellow line) and found that it matched the ISLI perfectly. It is important to note that a similar fit was observed in 60% of the oscillating neurons (n = 18) whereas in non-oscillating neurons (n = 70) the ISLI exhibited a Poisson-like distribution (*Figure 3H*). The strong correlation between oscillatory behavior and the occurrence of spikelets further supports the conclusion that these events represent activity in adjacent electrically coupled neurons.

## Estimating network connectivity from dual cell recordings of simultaneously occurring spikelets

*Figure 4A* depicts the spontaneous activity recorded simultaneously from two neurons. As described above (*Figure 3*) action potentials (diagonal bars in *Figure 4A*) occurred irregularly in either of the two neurons and were always associated with spikelets in the paired neuron (*Figure 4B*). The subthreshold activity was dominated by spikelets which appeared randomly in the two neurons. However, occasionally spikelets occurred simultaneously in both cells (marked in *Figure 4A* and shown at high resolution in *Figure 4C*) which we refer to as 'common spikelets'. Each of the three examples shown in *Figure 4C*, which occurred without measurable time difference, have variable amplitudes. The first and the third spikelets had larger amplitudes in the red neuron (cell 2) whereas the middle spikelet had a larger amplitude in the black neuron (cell 1). Since action potentials in one neuron evoke very similar spikelets in the other (*Figure 3A–B*), the most likely explanation is that each of these common spikelets represents the action potential in an additional neuron that is coupled to both of the recorded neurons (see Discussion). On the population level, 18 out of 30 pairs had common spikelets (60%). Of these pairs, the occurrence of common spikelets varied from 0.02 to 1.1 Hz, which is 3.5–66% of the total number of measurable spikelets (*Figure 4D*).

The occurrence of common spikelets can be used to estimate the number of neurons that are electrically coupled to each neuron in the olivary network. In this example of a paired recording, four different groups were identified (see Materials and methods, *Figure 4E*), which indicates that at least four neurons were electrically coupled to both recorded neurons. Further analysis of these data provided an estimate of the total number of neurons connected to each of the two recorded neurons. In this example, a total of 16 common spikelets, organized in four groups, were recorded. The four groups thus represent four neurons that are coupled to the two recorded neurons. Each of these neurons fired on average four times during the recording period. Therefore, we can assume that each neuron in the network also fired on average four times during the recording period. In addition to the common spikelets, nine non-common spikelets were recorded in the black cell and 32 in the red one. These non-common spikelets thus represent spikes in ~2 additional neurons (9/4) connected to the black neuron and eight neurons (32/4) connected to the red neuron. The result of this numerical consideration is that the black neuron is connected to the red neuron, to four additional neurons that are connected to both recorded neurons and to estimated two additional neurons, totaling seven neurons. Similarly, the red neuron is estimated to be connected to 13 neurons. This analysis was performed on 18 dual recordings and the results, which are summarized in

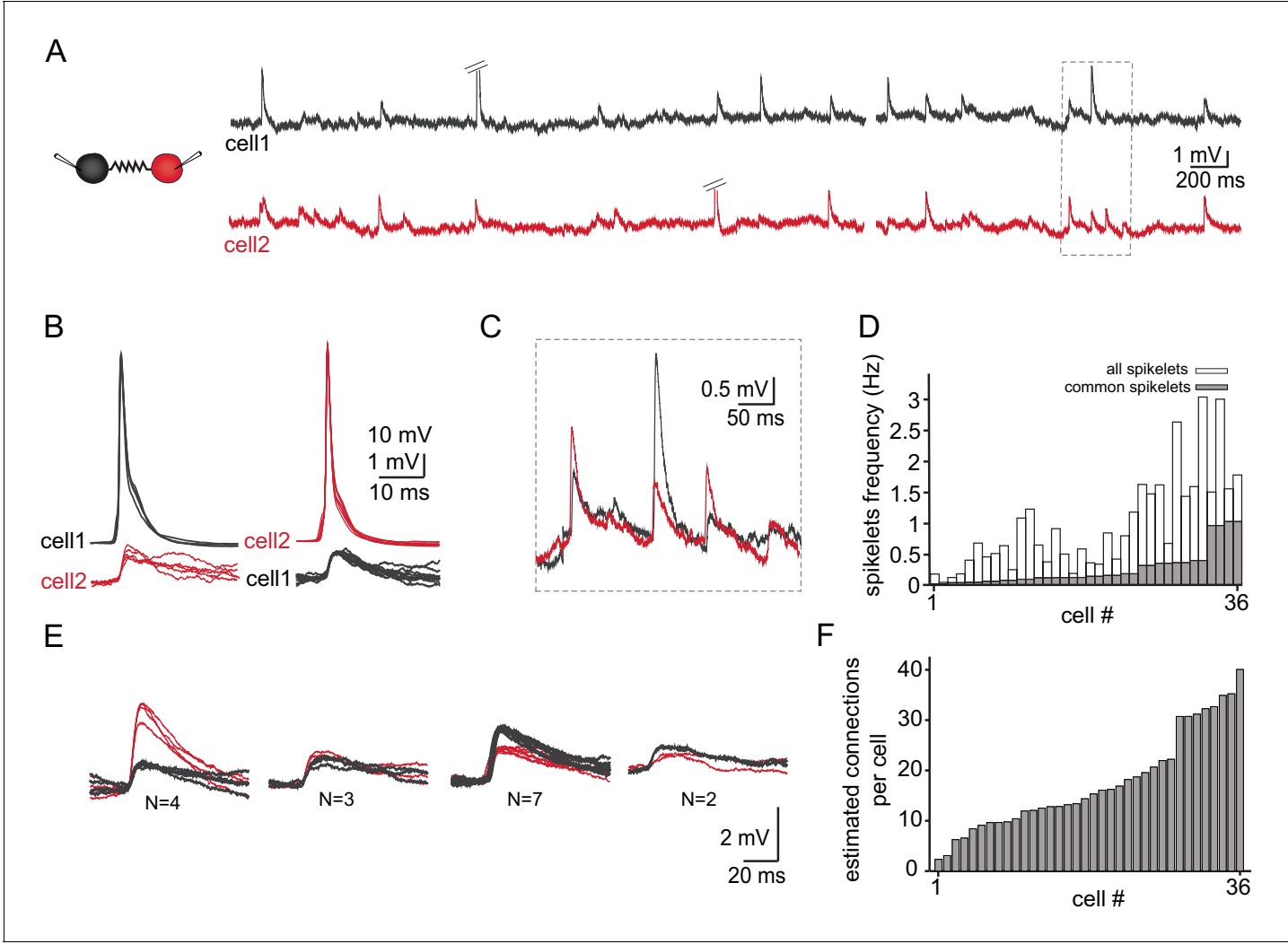

**Figure 4.** Common groups of spikelets during paired recording reveals the estimated number of neurons that are electrically coupled. (A) Simultaneous recording from two electrically coupled neurons. Action potentials were truncated (doubled diagonal lines) and an example of the occurrence of common spikelets is marked (dashed rectangle). (B) Superimposed traces of spontaneous action potentials in either cell 1 (black neuron) or cell 2 (red neuron) and the corresponding spikelets in the other neuron. (C) Higher magnification of the rectangle marked in A, showing spikelets that occur simultaneously in both neurons. (D) Histogram of the frequency of spikelets in neurons recorded in pairs, showing all the spikelets (white bars) and all the common spikelets (gray bars; n = 18 pairs), sorted by increasing frequency of common spikelets. (E) Example of common spikelets from the pair presented in A-C. The spikelets could be divided into four groups, with N = 2–7 spikelets in each group. (F) Histogram of the estimated number of neurons that are electrically coupled to each of the pair-recorded neurons (n = 18 pairs).

The online version of this article includes the following source data for figure 4:

**Source data 1.** Trace for *Figure 4*.

*Figure 4F*, indicate that a neuron can connect to as many as 40 other neurons (average of 19.2 ± 10.3). It should be noted that the use of a slice preparation undoubtedly contributed to the wide range of connected neurons and to some degree of underestimation (see Discussion).

We re-examined the approach to estimate the number of connections per neuron by reconstructing a realistic olivary network (*Figure 5A*, see Materials and methods). The firing rate of neurons in the network was set to 0.058 Hz ±0.04 Hz (as observed experimentally) and the number of common spikelets in pairs of neurons occurring within 15 min of simulation was measured. Recordings from a sample pair are shown *Figure 5B and C*. In this example, four groups of spikelets that appear 26, 16, 65 and 32 times were detected (*Figure 5C*). By applying the same calculation as performed in the experimental observations (*Figure 4*), we concluded that the red neuron was electrically connected to 24 neurons whereas the black neuron was connected to 26 neurons. In this model, the red

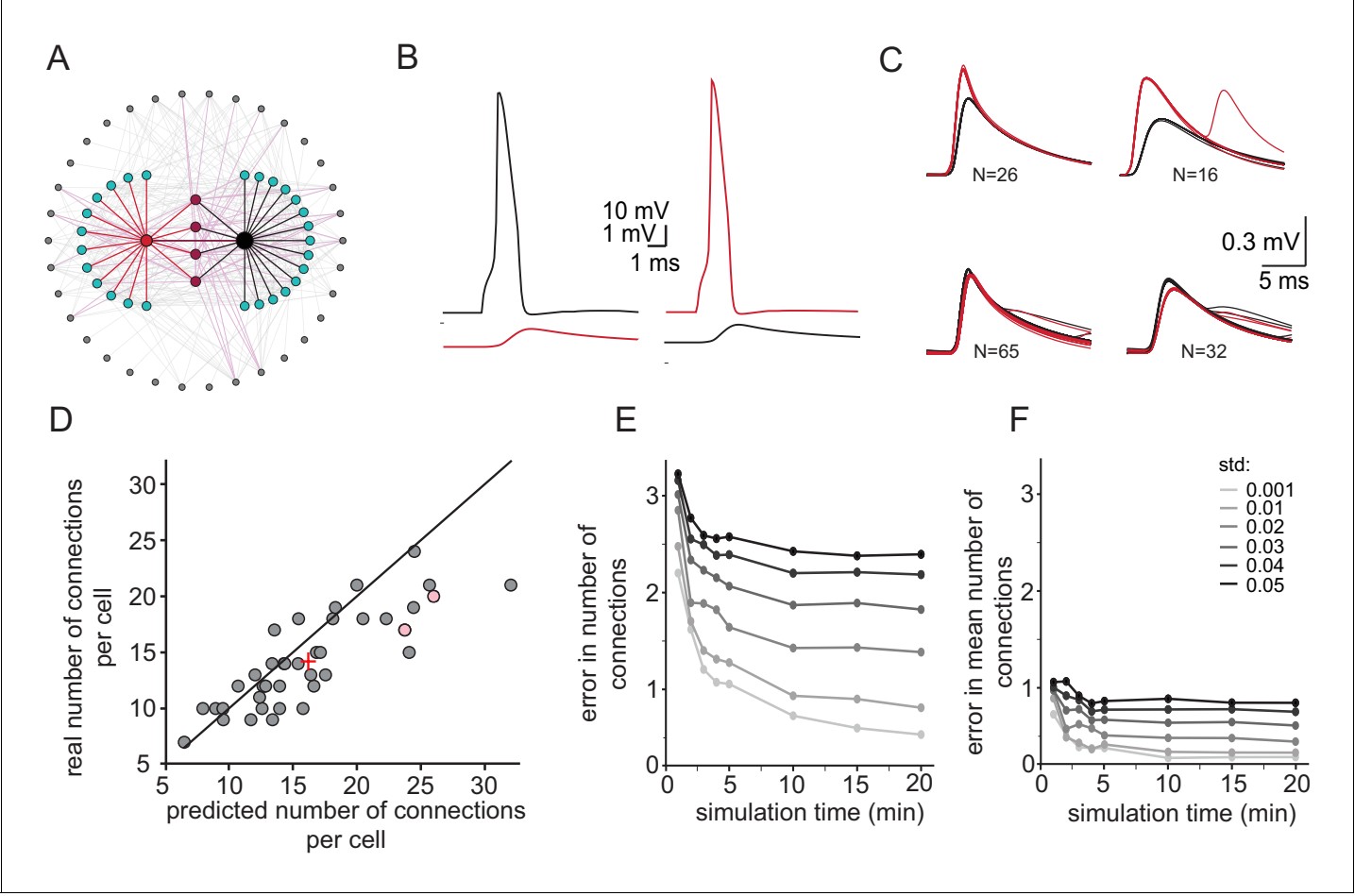

**Figure 5.** Simulations examining the method used for estimating the number of connections per cell. (A) Schematics of the modeled network where the recorded pairs of neurons (black and red circles) are connected to four common neurons (purple) and to 12 and 15 additional neurons (cyan); 34 other neurons (gray) that are connected to either the cyan or the purple neurons are also shown. (B) Example action potential and their post junctional responses from the red and black neurons in A. (C) The four common groups of spikelets recorded in the black and red cells with N = 16–65 spikelets in each group. (D) Plot of the predicted number of connections per cell, estimated from the common groups of spikelets, against the real number of connections per cell. The line marks the diagonal, the + sign marks the mean and the pink circles represent the two cells in A. (E) Difference between the estimated and real number of connections per neurons as a function of simulation duration for six different firing rate variabilities (std; color-coded as in the legend). (F) The difference in estimating the mean number of connections in the network (+ sign in D) as a function of simulation duration for six different firing rate variabilities (std). The calculation was done only on neurons that had common neighbors (n = 278 pairs). Mean firing rate was 0.058 Hz.

and black neurons were actually connected to 17 and 20 neurons, respectively (pink circles in *Figure 5D*). We performed the same calculation in 20 randomly selected pairs of neurons and plotted the estimated versus the real number of connections per cell (*Figure 5D*). The results were distributed along the diagonal (with the average marked by + sign), demonstrating the validity of this approach in estimating the number of connections for each neuron. However, the accuracy of the estimation depends strongly on variability in firing rate and the recording duration. As shown in *Figure 5E* the difference between the estimated and real number of connections per neuron decreases as a function of simulation duration. Moreover, at high variability in firing rate (black curve), longer recording duration does not improve the error compared to low variability firing rate (light gray curve). However, the estimation of the mean number of connected cells (+ sign in *Figure 5D*) is less sensitive to simulation duration or to variability in firing rate (*Figure 5F*, see discussion). We conclude that this approach provides a reliable estimate of the mean number of connected cells.

## Insights into the network architecture from the distribution of groups of common spikelets

Further insights into the organization of the network can be extracted from the distribution of the number of groups of common spikelets. *Figure 6A* depicts three examples of common groups of spikelets obtained from three different paired recordings, showing the reliability of the grouping procedure. The number of common groups (*Figure 6B*) varied from 0 to 7 with a markedly higher incidence at 2–4 groups. The distribution of the number of groups of common spikelets (*Figure 6B*) provides information on the organization of the network. This was examined by constructing artificial networks, each with different connectivity matrix, and calculating the expected distribution of common groups in the connectivity matrix. To that end, we first used experimental results showing that the probability to detect electrically coupled olivary neurons is distance dependent (dark green line in *Figure 6D*; *Devor and Yarom, 2002a*). After fitting these results with a Gaussian curve (*Figure 6D*, blue line) we constructed networks in which randomly selected neurons show the probability of connection as a function of inter-somatic distance fits the experimental distribution (*Figure 6D,G*; gray bars). Additionally, we checked the distribution of common neighbors (proxy for groups of common spikelets in *Figure 6B*) in the matrix (*Figure 6E,H*; gray bars) and compared it to the experimental distribution (*Figure 6E*, green line).

We examined two possible connectivity patterns that might support this type of distribution (see Materials and methods; network connectivity matrices). The first is a network where the probability of a connection between two neurons depends solely on their inter-somatic distance (*Figure 6C–E*). The second assumes that the network is organized into clusters of neurons (*Figure 6F–H*), where the probability of connection within a cluster is larger than between clusters (both probabilities are distance-dependent).

As shown in *Figure 6C–E*, the simple distance-dependent network captured the distance-dependent probability of a connection (*Figure 6D*, p=0.99, for detailed statistical analysis see Materials and methods and *Figure 6—figure supplement 1*), but it failed to reproduce the distribution of common spikelet groups as found experimentally (*Figure 6E*, p<0.002, Fisher's Exact). On the other hand, when the modeled network was organized in clusters, it replicated both the experimental distribution of common groups (*Figure 6H*, p=0.13) and the distance-dependent connection probability (*Figure 6G*, p=0.96). Note that in all modeled networks, each neuron was connected to about 11–21 neurons, which lay within the numbers estimated from the experimental observations (*Figure 4F*).

To further investigate the robustness of the result that the IO network is organized in clusters of coupled neurons, we searched for possible experimental or computational artifacts that may affect this conclusion. First, we examined the possibility that the experimental observation of the distribution of common groups is inaccurate. To that end, we tested the possibilities that we either failed to detect common groups or incorrectly identified groups of common spikelets.

Accordingly, we increased the number of pairs that did not show common groups to 18 (6 additional pairs) and removed one pair from each of the other groups (*Figure 7A*, green line). Alternatively, we reduced the number of pairs that did not show common groups by half (six pairs) and distributed them among the other pairs (*Figure 7B*, green line). As shown in *Figure 7A and B*, these variations did not change the main conclusion, namely the non-cluster organization cannot account for the distribution of the common groups (p<$10^{-6}$ for *Figure 7A* and p=0.0025 for *Figure 7B*; see Materials and methods).

Furthermore, we also examined other cluster connectivity profiles (see Materials and methods). *Figure 7C and D* demonstrate two additional examples of clustered organizations (gray and black bars) that differ in their maximal connectivity value ($\Sigma$) and the reduction of probability of connection as a function of distance both for within and between clusters ($\sigma$, see inset for connectivity profiles). Both models faithfully reproduced the distance-dependent probability of a connection (*Figure 7C*; p=0.7 and p=0.75 for gray and black, respectively) and the distribution of common groups (*Figure 7D*, p=0.9 and p=0.67 for gray and black, respectively). On the other hand, under non-clustered organization, changing the maximal connectivity value ($\Sigma$) or the reduction of probability of connection as a function of distance ($\sigma$, color coded) did not reproduce the observed distribution of common groups (*Figure 7E–G*). We scanned different $\sigma$, $\Sigma$ and show that all networks produce a

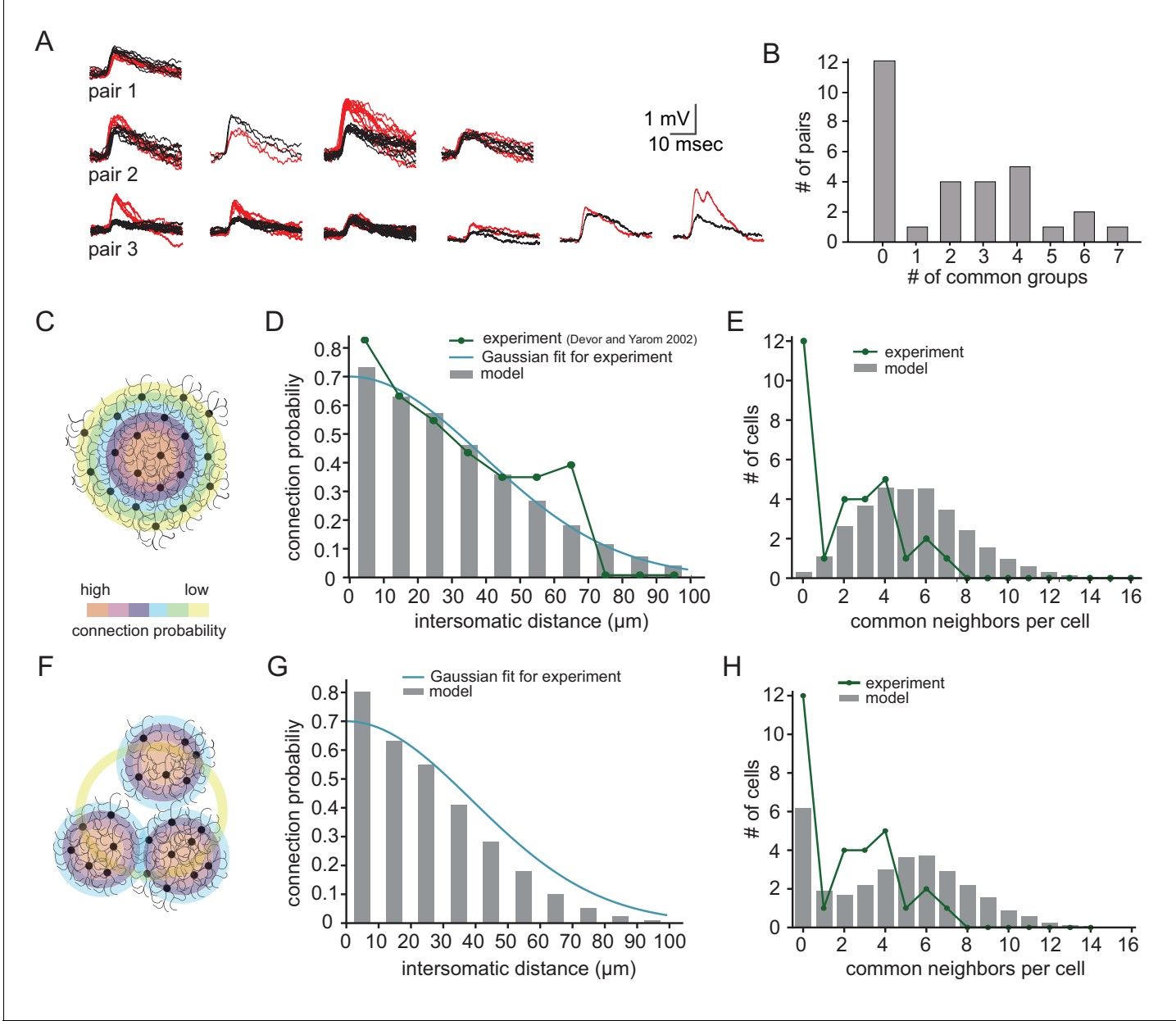

**Figure 6.** Simulations examining the network connectivity that accounts for the experimental distribution of common groups of spikelets. (A) Experimental example of common spikelets from three different pairs, that were divided to 1, 4 or 6 common groups (pair 1,2 and 3, respectively). (B) The distribution of the number of common groups in all experimentally recorded pairs. (C–E) The expected distribution of the common groups in a model where the probability of connection is distance- dependent. (C) Schematic illustration of a distance-dependent connectivity. The connection probability is color coded. (D) The probability of connection in the model (gray bars) and in the experiments (blue line) as a function of the inter-somatic distance. The blue curve represents a Gaussian fit to the data. The green curve represents the experimental results (*Devor and Yarom, 2002a*; see actual data in *Figure 6—figure supplement 1*). (E) Distribution of the common groups in the model (gray bars) and experiment (as in B; green line) for cells of up to 40 μm apart. (F–H) Same as C-E for a network that is organized in clusters of neurons with a high probability of connection within a cluster and a low probability between clusters (See Materials and methods). Each cluster consisted of about 40 neurons.

The online version of this article includes the following source data and figure supplement(s) for figure 6:

**Source data 1.** Traces for *Figure 6*.

**Figure supplement 1.** Extended data for the statistical analysis in *Figure 6*.

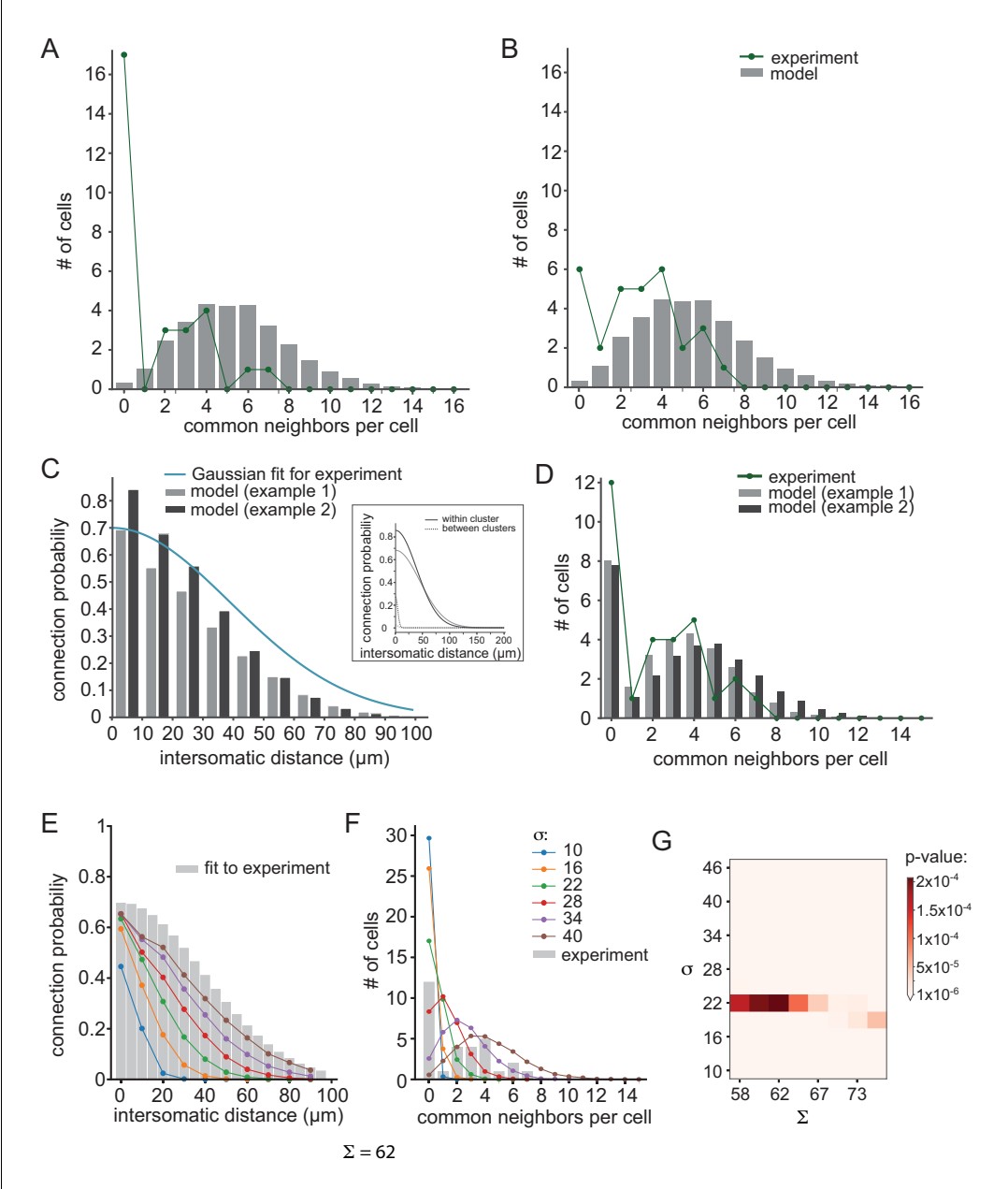

**Figure 7.** Further examination of possible network connectivity reveals the robustness of cluster organization as the best explanation for the experimental data. (A) and (B) Examining the possibility that inaccurate detection of common groups led to the clustered organization. (A) Compensating for possible over estimation of common groups, by increasing the number of pairs that had zero common groups. (B) Compensating for possible underestimation of common groups, by reducing the number of pairs that has zero common group. Green lines are the corrected distribution of common group; bars are the expected distribution assuming distance-dependent connectivity (As in *Figure 6C–E*). (C–D) Two additional cluster models that can account for the connectivity probability and common neighbor distribution that were found experimentally (as in *Figure 6F–H*). The different connectivity profiles of the models are shown in the inset. (E–G) Color lines show the distance- dependent connection probability (E) and common neighbor distribution (F) for seven different distance- dependent models with $\Sigma = 62$ and different $\sigma$ (see legend). All models did not fit the experimental common neighbor distribution (p<0.05). (G) Summary of the fit between the experimental common neighbor distribution and modeled common neighbor distribution, for different distance dependent models. The x axis represents the $\Sigma$ that was used for each network and the y axis represent the $\sigma$ that was used. All p values were below 0.05 (see color bar). The networks in E and F correspond to the column marked by $\Sigma = 62$.

distribution of common neighbors that is significantly different than the experimental result (*Figure 7G*).

Thus, we show that the IO is not connected with a simple distance-dependent rule; instead it is more likely that it is organized into clusters of neurons with a higher probability of connection within clusters and a low connection probability between clusters.

## Discussion

In this study, we measured subthreshold unitary activity from neurons of the inferior olive in slice preparation. The implementation of a variety of experimental approaches linked to computational simulation led to several important conclusions regarding the organization of the network of electrically coupled neurons in the inferior olive nucleus. We showed that there are two populations of unitary events that differ in their waveform and amplitude. The spikelets represent the occurrence of action potentials in a coupled neuron, while the fast events are likely to represent intrinsic regenerative responses at remote locations. It should be noted that in the scientific literature the word 'spikelet' refers to all non-synaptic subthreshold unitary events and thus creates a certain lack of consistency regarding their origin (see Introduction; *Michalikova et al., 2019*). The uniqueness of our experimental system lies in its ability to differentiate between two types of non-synaptic unitary events and thus to characterize them. We then used the spikelets recorded simultaneously from two neurons to gain insights into the size and organization of the electrically coupled network within the IO nucleus. Analysis of the experimental results showed that each olivary neuron is connected to ~20 other neurons and theoretical considerations indicated that the neurons are not connected in a simple distance-dependent manner, rather that the network is organized into clusters of neurons, where the probability of connection within a cluster is higher than the probability of connection between clusters.

### The electrically coupled network in the olivary nucleus

It is well-established that the olivary nucleus forms an electrically coupled network. It has been suggested that this network operates as a synchronous rhythmic device, capable of generating precise temporal patterns (*Jacobson et al., 2008*). Both synchronicity and rhythmicity are generated by the delicate interplay between electrical coupling and ionic conductances. Thus, a single cell by itself cannot oscillate, whereas in a network formation the neurons generate subthreshold oscillations (*Manor et al., 1997*; *Loewenstein et al., 2001*; *Chorev et al., 2007*). In this work, we studied the relationship between spikelets and subthreshold oscillatory activity and found that in oscillating cells the occurrence of spikelets coincided with the depolarizing phase of the oscillation whereas in non-oscillating cells they seemed to be randomly distributed (*Figure 3G,H*). This result strongly supports our previous hypothesis that the occurrence of subthreshold oscillations is a network phenomenon. Therefore, when the recorded cell is oscillating, the entire network is synchronously oscillating, thus generating action potentials at the peak of the oscillation that appear in the recorded cell as spikelets. Theoretically, by calculating the number of spikelets at the peak of the oscillatory activity, one should be able to calculate the number of coupled neurons in the oscillating network. Although tempting, this is practically impossible because spikelets, given their small amplitude and noisy environment, cannot be classified into groups. Therefore, we used the common spikelets to estimate the number of coupled neurons.

Recording from two olivary neurons revealed spikelets that occurred simultaneously in both recorded neurons. Given that the average rate of spikelets is $0.56 \pm 0.62$ Hz, the probability that these common spikelets reflect random occurrence is extremely low. Furthermore, the repeated appearance of common spikelets with the same relationships (amplitude ratio and waveforms) further supports the non-random occurrence of these events. Thus, the accurate timing of common spikelets can only be attributed to a common source; that is a single pre-junctional neuron. The number of neurons that were coupled to the two recorded neurons, which varied from 1 to 7, should be correlated with the size of the network; more common spikelets are expected in larger interconnected networks.

Our simple method of calculating the size of a coupled network is based on data obtained during simultaneous recordings from two neurons and on the assumption that the neurons display a similar rate of spontaneous spiking activity. Our simulations show that the accuracy of this method is mainly

affected by variability in the firing rate (*Figure 5E,F*). To minimize the error in estimating the firing rate of the neurons in the network, we used only the spontaneous rate of the common spikelets, and not the firing rates in the recorded neurons which are affected by the intracellular recordings. Under this assumption, we showed that neurons are connected to 3–40 other neurons. This is in line with other studies reporting 1–38 (*Hoge et al., 2011*) or 0–33 (*Placantonakis et al., 2006*) dye-coupled neurons. This broad variability can be attributed to the use of an in- vitro system, where differences in the number of cells and the integrity of the circuit are characteristic features. Alternatively, this large variability might reflect an innate feature of the IO nucleus where electrical coupling is under continuous modulation (*Lefler et al., 2014*; *Mathy et al., 2014*; *Turecek et al., 2014*).

In addition to calculating the number of connected cells, the distribution of common spikelets enabled us to study the connectivity profile within the nucleus. Our data showed that each two neurons had 1–7 common groups. However, there were a large number of paired recordings that failed to show common groups. Using a theoretical approach, we demonstrated that this distribution should not be expected if we assume that the probability of connection depends solely on the distance between the neurons. On the other hand, if the nucleus is organized into clusters where the probability of connection within the cluster is higher than between clusters, the observed distribution of common groups can be reproduced. Although the size of the clusters, as well as the probability of connection cannot be defined with the current data, this constitutes the first physiological study that supports the assumption of clustered organization of the nucleus deduced mainly from morphological studies.

This approach to analyze electrically coupled networks should not be restricted to the inferior olive network. Most of the studies on electrical coupling in central neurons used paired recordings, an essential procedure to demonstrate electrical coupling. In many of these recordings spontaneous spikelets are readily observed. In the work of *Curti et al. (2012)* on the Mesencephalic Trigeminal Nucleus, the occurrence of a spikelet in only one of the neurons is taken as indication for additionally connected cell. Similarly, the work of *Long et al. (2004)* on the Thalamic Reticular nucleus demonstrates spontaneously occurring spikelets of different sizes in both recorded neurons. Thus, to implement our approach all one needs is paired recordings of spontaneous activity of cells in electrically coupled network.

In summary, we presented a comprehensive study that implemented a wide range of research approaches to unravel the functional architecture of the inferior olivary network. We showed that new insights into the organization of the network can be gained by analyzing spontaneous subthreshold events, thus paving the way for a novel experimental and theoretical approach to the study of electrically coupled networks.

# Materials and methods

## Animals
All experimental procedures were approved by the Hebrew University's Animal Care and Use Committee. Brain stem slices were prepared from the following strains of mice: C57BL/6, B6.Cg-Tg (Thy1-COP4/EYFP; Jackson Laboratory) and Gad2-tm2(cre)Zjh/J (Jackson Laboratory).

## Slice preparation
Mice were anesthetized with an intraperitoneal injection of Pentobarbital (60 mg/Kg), and 300 µm coronal brainstem slices containing the inferior olive were dissected using a Campden 700smz slicer (Campden Instruments), in 35°C physiological solution containing 126 mM NaCl, 3 mM KCl, 1.3 mM $MgSO_4$, 1.2 mM $KH_2PO_4$, 26 mM $NaHCO_3$, 10 mM glucose, and 2.4 mM $CaCl_2$, gassed with 95% $O_2$ and 5% $CO_2$. Slices were left in physiological solution at 35°C for 0.5–8 hr until transferred to the recording chamber.

## Electrophysiological recordings and ChR stimulation
The recording chamber was perfused with 95% $O_2$ and 5% $CO_2$ physiological solution at 24–28°C. Slices were visualized using a 40X water-immersion objective in an Olympus BX61WIF microscope equipped with infrared differential interference contrast (DIC). In order to record from intact olivary networks, recordings were targeted to the deepest neurons possible in the slice. For pair recordings,

two cells located up to 50 μm apart were selected. Whole-cell recordings were performed using 6–9 MΩ glass pipettes with intracellular solution containing 4 mM NaCl, $10^{-3}$ mM CaCl$_2$, 140 mM K-gluconate, $10^{-2}$ mM EGTA, 4 mM Mg-ATP, and 10 mM HEPES (pH 7.2). Signals were acquired at 10–20 KHz using a Multiclamp 700B (Molecular Devices) and LabView-based custom-made acquisition software (National Instruments and ZerLabs). For the ChR experiments in Thy1 mice, a custom-made digital mirror light stimulator with a LED light source (460 nm; Prizmatics) was used to activate the ChR at defined locations on the slice. In some experiments either TEA (10 mM), DNQX (20–40 μM) or DNQX (40–60 μM) and AP-5 (40–100 μM) were added to the recording solution.

## Data analysis and statistics

Analysis was performed using MATLAB (R2014b and R2016a, MathWorks) for the experimental data and Python 2.7 for the simulation data. The 70 neurons that were selected for detailed analysis had a frequency of subthreshold events exceeding 0.02 Hz. The events were divided into two different groups according to their amplitude and rise time. The event rise time was calculated as 10–90% of the amplitude. The fast event groups were clustered using the K-means clustering method, using the MATLAB 'evalclusters' function. The effect of the DC current injection in *Figure 2C–H* was measured in 16 neurons for different values of current injection. For each neuron, the average value for each current injection was calculated (gray dots in *Figure 2C–H*) and fitted with a linear line (dashed gray lines). To calculate the average slope (black lines), we averaged the gray dots for each current injection (black dots) and fitted them with a linear line. Error bars represent STD. A one-sample t-test was used to compare the distribution of the slopes of the linear fits of each cell (dashed gray lines) to a distribution with a mean equal to zero. A paired t-test was used to test for differences in the effect on spikelets and fast events. The frequency values for spikelets and fast events (*Figure 2G–H*) were normalized for each cell to the highest value. The normalized fast events frequency (*Figure 2H*) was calculated from both fast events and the action potentials that were evoked from the fast events.

The coupling coefficient (CC, *Figure 3C*) was calculated as the ratio between the change in the steady-state voltage of the post-junctional cell and that of the pre-junctional cell in response to 250 ms current injection in the pre-junctional cell. The spike coupling coefficient was measured as the amplitude of the post-junctional spikelet divided by the amplitude of the pre-junctional action potential. A Pearson correlation was used to calculate the p-value of the linear regression in *Figure 3C*. To detect spikelets in oscillatory traces, the raw trace was subtracted with a low-pass filtered trace. The Inter-spikelet-interval (ISLI) in oscillating neurons (*Figure 3E*) was only calculated in neurons that had more than 150 spikelets during the session. The ISLI histogram was computed using 4 ms time bins, and the autocorrelation in oscillating neurons was calculated using a lag of 1 ms.

Common spikelets were defined as spikelets that were detected in a paired recording in both cells simultaneously. To that end, we searched for spikelets which peaks occurred in both cells within a time window of 8 msec. Groups of common spikelets in the two cells were clustered according to their amplitudes using k-means analysis. These clusters were then grouped according to the ratios between amplitudes in the two cells and verified manually. To estimate the number of connections per cell, the total number of spikelets ($T_{spikelets}$) was multiplied by the number of groups ($N_{groups}$) and divided by the number of common spikelets ($C_{spikelets}$):

$$\mathrm{Estimated\,connections\,per\,cell} = \frac{T_{spikelets} * N_{groups}}{C_{spikelets}}$$

If the two recorded cells were coupled (i.e., a spike in one cell gave rise to a spikelet in the other cell), +one was added to the estimation of connections for these two cells.

## Neuron models

In a few experiments (using C57BL/6 mice), Neurobiotin (0.5%; Vector Laboratories) was added to the pipette solution to label the recorded neurons. Slices were then fixed in 4% paraformaldehyde overnight, washed in PBS and stained with 1 μg/ml Streptavidin AlexaFluor 488 (Life Technologies). Using the Neurolucida software (MBF Bioscience), three olivary neurons were reconstructed from fluorescence image stacks acquired using a Leica TCS SP5 confocal microscope (Leica Microsystems).

To compensate for tissue shrinkage, the z-axis of the reconstruction was multiplied by a factor of 3. A compartmental model was generated from the morphological reconstruction using NEURON (**Carnevale and Hines, 2006**). The axial resistance ($R_a$) was set to 100 $\Omega$cm, the specific membrane capacitance ($C_m$) to 1 $\mu$F/cm$^2$ and the specific membrane resistivity ($R_m$) for the three reconstructed cells were 4300, 4500, 3800 $\Omega$cm$^2$ respectively. These values were chosen to yield an input resistance ($R_{in}$) that was within the experimental range (115 ± 43 M$\Omega$).

## Building the IO network connectivity matrices

We constructed a network of IO composed of 1134 neurons randomly distributed within a volume of 250 × 500×200 $\mu$m, which resulted in 0.045 neurons per 10 $\mu$m$^3$. We then clustered the neurons by their location using k-means clustering, and varied the number of neurons in a cluster by choosing k to be 1134 divided by the number of neurons in a cluster. The probability of a connection between two neurons decays with distance according to a Gaussian profile:

$$\frac{\Sigma * e^{-\frac{x^2}{2*\sigma^2}}}{100}$$

where $\Sigma$ is the maximal probability for connection (when the distance between the neurons is 0), x is the distance between neurons and $\sigma$ sets the decay of connection probability with distance (see **Figure 7C** inset for examples). Note that the shape profile of neuron connectivity within a cluster could have a different $\Sigma$ and $\sigma$ than the connectivity profile of neurons belonging to different clusters. The common neighbor distribution (**Figure 6F,I**) was extracted on randomly selected pairs of neurons within a distance of 40 $\mu$m from the connectivity matrix. The network shown in **Figure 6G and H** had $\Sigma$ and $\sigma$ of 77 and 45 within a cluster and 15 and 20 between clusters, respectively. The networks shown in **Figure 7C and D** had $\Sigma$ and $\sigma$ within cluster of 68.1, 44.3 (gray model) and 85.6, 38.5 (black model), respectively. And $\Sigma$, $\sigma$ between clusters of 2.4, 4.5 (gray model) and 27.3, 4 (black model), respectively. Number of neurons in a cluster was set to 40 in all cases.

## Constructing the IO network model

To simulate a realistic network of IO neurons (**Figure 5**), we followed the steps described above but with a few modifications. The network volume was 125 × 250×100 $\mu$m, and populated with 180 neurons (0.057 neurons for 10 $\mu$m$^3$). These neurons were cloned from the three 3D-reconstructed olivary neurons. $\Sigma$ and $\sigma$ within cluster were 77 and 45, respectively; and $\Sigma$, $\sigma$ between clusters were 15 and 20, respectively (as in **Figure 6G,H**). The electrical connection between two neurons was mediated by two gap junctions (GJs). A GJ conductance (GJc) of 0.3 nS resulted in a coupling coefficient of 0.03 ± 0.019 as in the experimental range (0.039 ± 0.029). After adding GJc to the modeled cell, $R_m$ was modified to maintain the experimental value of $R_{in}$ (see details in **Amsalem et al., 2016**). The spikes in the networks were created by current injection to the soma (simulated spikes) following a Poisson process. We ran the network for 15 seconds with dt of 0.025 ms, we automatically detected the spikelets and clustered them as done experimently, but using bd-scan instead of k-means. **Figure 5E, F** represent the best-case scenario, assuming all spikelets were detected and clustered correctly.

## Statistical comparison between the model prediction and the experimental observation

The experimental observations were compared with the models' predictions on two levels. One is the probability to get the distribution of common group and the second is the probability of connection as a function of distance. To compare the distribution of common groups (**Figures 6–7**), we used the Monte Carlo Fisher's Exact method (**Noutahi, 2018**) with 100,000 replicates (see example in **Figure 6—figure supplement 1C**).

To compare the distribution of probability for connection as a function of distance, for each network configuration we tested the connectivity of pairs sampled from the model and used two statistical methods to compare the experimental sample to the sample from the model. In the first method we constructed a contingency table from the model and the experimental data, and for each distance calculate the p-value using Fisher's Exact (2 × 2, using Python SciPy), we then merged those 10 p-values with Fisher's combined probability test.

In a second method, for every distance we normalized the expected data and multiplied by the number of sampled in the experimental data for this distance, providing a vector of expected observation (see example in *Figure 6—figure supplement 1A,B*). We used Chi-square (using Python SciPy) to compare between these expected values to the observed values. (In order to get five samples per cell, we merged cells 60–70, 70–80, 80–90 and 90–100 in the connected column, and cells 80–90 with 90–100 in the non- connected column). Both statistical methods resulted with comparable p- values. The p- values presented in the main text are for the Fisher's Exact method.

## Acknowledgements

This study was supported by the Israel Science Foundation (http://www.isf.org.il/) grant #1496_2016 and by the Einstein Foundation Berlin. IS and OA were supported by grant agreement no. 604102 'Human Brain Project', a collaborative grant under the Blue Brain Project and a grant from the Gatsby Charitable Foundation. YL was supported by a postdoctoral fellowship from the Edmund and Lily Safra Center for Brain Sciences (ELSC). We wish to thank Prof. Israel Nelken for statistical advice.

## Additional information

### Funding

| Funder | Grant reference number | Author |
|---|---|---|
| Israel Science Foundation | #1496_2016 | Idan Segev Yosef Yarom |
| Gatsby Charitable Foundation | | Idan Segev |
| Human Brain Project | 604102 | Idan Segev |
| Edmund and Lily Safra Center for Brain Sciences | | Yaara Lefler |
| Einstein Foundation Berlin | | Yosef Yarom |

The funders had no role in study design, data collection and interpretation, or the decision to submit the work for publication.

### Author contributions

Yaara Lefler, Data curation, Formal analysis, Investigation, Visualization, Methodology; Oren Amsalem, Data curation, Software, Formal analysis, Investigation, Visualization, Methodology; Nora Vrieler, Data curation; Idan Segev, Supervision, Funding acquisition; Yosef Yarom, Conceptualization, Supervision, Funding acquisition

### Author ORCIDs

Yaara Lefler  https://orcid.org/0000-0001-8911-7034
Oren Amsalem  https://orcid.org/0000-0002-8070-0378
Idan Segev  http://orcid.org/0000-0001-7279-9630

### Ethics

Animal experimentation: All experimental procedures were approved by the Hebrew University's Animal Care and Use Committee. The Hebrew University is an Association for Assessment and Accreditation of Laboratory Animal Care (AAALAC)-accredited institution #12005.

### Decision letter and Author response

Decision letter https://doi.org/10.7554/eLife.43560.sa1
Author response https://doi.org/10.7554/eLife.43560.sa2

## Additional files

### Supplementary files
• Transparent reporting form

### Data availability
All source data for this article is provided in the source data files associated with the figures.

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
