## [Decision Letter]

**Acceptance summary:**

This paper provides novel insights on the properties and organization of neural networks in the inferior olive, and electrically coupled networks in the vertebrate CNS in general. In contrast to chemical synapses, networks formed by electrical coupling are more difficult to analyze because of their characteristic bidirectionality of transmission. Inferior olivary neurons are connected only by electrical coupling, thus providing an ideal structure to explore networks primarily organized by electrical coupling. The analysis presented makes the convincing argument that, rather than forming a single 'matrix' of electrically coupled cells, there exist clusters of functionally interconnected (electrically coupled) neurons. The authors arrive to this conclusion by disambiguating the nature of subthreshold responses and analyzing those representing coupling of action potentials, a step that allows them to estimate of the number of coupled cells with fair accuracy. These estimates were supported by computer simulations which provided novel insights into the organization of the network. The experiments are elegantly designed, and the paper is well illustrated. One of the important features of the work is that it presents a physiological framework for analyzing the functional architecture of other electrically coupled networks.

**Decision letter after peer review:**

Thank you for submitting your article "Using subthreshold events to characterize the functional architecture of electrically coupled networks" for consideration by *eLife*. Your article has been reviewed by three peer reviewers, and the evaluation has been overseen by Ronald Calabrese as the Senior and Reviewing Editor. The following individuals involved in review of your submission have agreed to reveal their identity: Alberto Pereda (Reviewer #1); Chris I De Zeeuw (Reviewer #2).

The reviewers have discussed the reviews with one another and the Reviewing Editor has drafted this decision to help you prepare a revised submission.

Summary:

The authors have studied subthreshold membrane fluctuations of inferior olivary (IO) neurons in vitro. They observed two distinct types of events: fast, relatively large "Internal Regenerative Events (IREs)" that they identify as intrinsically generated regenerative events, and slower, smaller "spikelets" that they identify as electrical synapse-mediated events. Using various electrophysiological and computational lines of evidence they conclude that the IREs are likely initiated at axonal spines near the somata of IO cells (a significant novel finding), while spikelets are electrical synapse (gap junction)-mediated events generated by presynaptic action potentials. The authors also use spikelet data to estimate the size and pattern of electrically coupled neuronal clusters.

Essential revisions:

The experiments and modeling are well done, the analyses are interesting, and the conclusions of the study are potentially very important. There are several concerns about the main findings – the origins of the IRE and the network (neuronal cluster) analyses that must be addressed in revision. The detailed comments of the reviewers amplify these major required revisions described below; the focus in revision should be on these major points and not on minor differences in the reviewer comments.

1) All reviewers agree the synaptic pharmacology needs to be better both for the IRE and spikelet stories. The authors state they tested 19 cells in CNQX/APV, so perhaps all that's left is to do more detailed analysis and document it in the paper.

2) Are axonal spines really as common and beautiful in mouse cells as De Zeeuw showed them to be in the cat? The authors have already dye-filled mouse IO cells, but didn't show them in the paper. These spines should be documented in the paper (LM level) and perhaps these data are already available.

3) Can the authors provide more direct evidence that the IREs are coming from spines? Modeling is not enough, especially when we don't even know if axonal spines exist. Most people believe spikes initiate at axon initial segments because of high Na channel density there, and there are nice markers for initial segment-specific protein complexes (e.g. immuno for ankyrin-G, or Na channels themselves). Such a demonstration in axonal spines (LM level) by itself would be enough. Better electrophysiology (glutamate uncaging, axonal recording) would also be great, but probably too time-consuming for revision.

4) Regarding the functional significance of the IREs, the authors never mention whether those big subthreshold IREs trigger electrotonically propagated events (small spikelets?) in coupled cells. These are data the authors should already have from their paired recordings. In general, it's not at all clear how IREs, spikelets, spontaneous oscillations, and full spikes interact in coupled networks of IO neurons; the authors should present a more holistic view of how all these membrane events interact.

5) The other major and potentially interesting but problematic part of the paper is about spikelets, coupling, and clustering. The authors might be able to strengthen this without doing more experiments, if they can address some of the technical issues raised in the reviews (about clarity of description, noise and detection thresholds, alternative connection schemes in the model, statistical reliability of modeling curve fits, etc).

Reviewer #1:

There are several concerns:

– The paper has two sections, the analysis of the subthreshold responses and the use of spikelets to analyze network connectivity. While the first section reads very well I had problems with the transition to network analysis. I found the two sentences at the start of paragraph two of subsection “Estimating network architecture from dual cell recordings of simultaneously occurring spikelets” very confusing. I had to read them several times. There is little information on how common spikelet amplitude and ratio is used to generate the groups. Although I understand the notion, the details still unclear to me. The criteria and usefulness of these values for grouping spikelets should be explicit and not let the reader guessing.

– Coupling between inferior olivary neurons is heterogenous and one cell can be differentially coupled to its various neighbors. I am not sure how this would impact the analysis performed by the authors, in particular the ratio between common spikelets. Was coupling heterogeneity incorporated into computer simulations? The authors should address this issue.

– I suggest making the title specific to the inferior olive, perhaps ‘Using subthreshold events to characterize the functional architecture of *the inferior olive’*. While the number of coupled cells was not a surprise and consistent with previous reports using different methodology, the analysis performed by the authors provides further insights on the organization of the networks (favors the existence of compartments of functionally connected neurons) and this should be reflected in the title. In addition, although theoretically possible, it is unclear if the method could be used to analyze other, less known, networks. Are the authors suggesting the method could be applied other networks? If so, they should specifically discuss this point in the Discussion section.

Reviewer #2:

I have several suggestions to improve the manuscript:

– The spike-rates of the IO cells were approximated, and thus a possible source of bias. Though the authors do say they estimated the firing rates based on spontaneous slow event-rates, and not on the spiking rate of the recorded cells as these are affected by the intracellular solution, I wonder to what extent the estimates of the variability and frequency of the cells' firing rates result from the cell-attached recordings.

– With regard to the source of the fast spontaneous events, or "Internally Regenerative Events (IREs)", the authors find that these IREs do not depend on gap-junctions, but originate from spines in the axon hillock of the cell. I find this hypothesis quite plausible and attractive, but I wonder whether some additional electrophysiological experiments can be provided, either with glutamate uncaging or maybe glutamate or AMPA application with a secondary "puffing" pipette. This might confirm the effects of these "overruling" axonal spines on the IO neurons. More specifically, it would be nice to see an experiment with direct and/or solitary stimulation of the axonal spines.

– The authors suggest that the amplitudes of the IREs are predominantly determined by the neck length. However, I wonder why the observed amplitudes are not normally distributed but distributed in a sinusoid shape (see Figure 1B). I would assume that the spine neck lengths would be distributed normally. If the assumption is that the higher amplitudes are the result of summation, then I wonder why the sinusoid is so clean (because variable neck length would reduce the precision of the sinusoid shape, as the individual events that make up the summated event would be more variable as well). Additionally, as is mentioned in the text, De Zeeuw et al., 1990 showed that up to 8 spines emerge from the axon initial segment. Would the authors not expect a larger spread of amplitudes if that were the case?

– Given that the slow spikelets predominantly occur on top of the STOs, given that action potentials in one cell often lead to spikelets in electronically coupled cells, and given that IREs often induce action potentials, wouldn't the authors expect more spikelets outside of the normal STO-spikelet-window? Or do the authors assume that the afferents to the axonal spines are somehow also influenced by the STOs? Or do the effects of the IREs depend on the phase of the subthreshold oscillations? If that were the case, would the functionality of an emergent action-potential inducer that foregoes dendritic input not be somewhat hampered?

– There is a dependency between the underlying oscillations and the number of slow spikelets. However, spikelets would be harder to detect in steep rising phases of the oscillation. How did the authors deal with this potential confounder?

– The distribution of data about amplitudes and timing is probably not normal, so it might be better to report median and quartiles rather than mean and standard deviations throughout the article.

– I miss a plot of the distribution of STO amplitudes, particularly in the context of the claim that gap junctions are necessary for spontaneous oscillations. Our current understanding of the cell mechanisms generating the oscillations suggests that STO's may be generated in a few high amplitude cells, with other cells either reporting that oscillation, or indeed, engaging in them. Possibly in a network in which cells are connected by gap junctions a combination of these two phenomena exists. In fact Leznik and Llinas, 2004, showed that STO's are maintained within single cells following gap junction blocking, though the network synchrony is disrupted. So in my view, it is a combination in which the level of coupling will contribute to the level of STO's, while the coupling is not absolutely essential.

– The manuscript gives the impression that cells are either oscillating or not, while the literature has reported intermediate cells 'conditional oscillators'. This is an important point, since dynamical system's analysis shows that a non-oscillating cell can promptly be led to oscillate upon perturbations (Schweighofer et al., 1999).

– While it is likely that oscillations are potentiated by coupling, 'network oscillations' of a couple of mV's could simply be a read-out of high amplitude oscillators. Please discuss this point more extensively.

– It would have been nice to see simulation runs with two alternate connectivity scenarios. One with fewer connections, the other with more – the latter reflecting the possible scenario in the intact preparation.

Reviewer #3:

The experiments and modeling were well done, the analyses are interesting, and the conclusions of the study are potentially important. If small spike-like events actually originate from spine-like structures on the axon initial segment, it would be a novel and unique observation as far as I know. If electrical synapse-generated spikelets can be used to reliably estimate the size and patterns of gap junction-coupled neuron clusters, that would be an important contribution to IO neurobiology and perhaps more widely. As presented here, however, I don't think either conclusion is strongly supported by the evidence.

1) This study rests entirely on the properties of small, subthreshold events, so clear reporting of their characteristics is critical. A first-order question is whether any of these events represent chemical synaptic input. Are they EPSPs? Are they triggered by EPSPs? The answers here are vague. The authors report that the application of CNQX reduced the rate of "unitary unipolar events" from an average of 0.7 Hz to "below 0.02 Hz" (Results paragraph one), but no details are provided. This control rate was apparently taken from a large group of cells, not from the 19 cells to which the drug was applied. The authors should report the data properly, with measurements from the same neurons before and after CNQX. They should also report the characteristics of the subthreshold events remaining after CNQX, including amplitudes and kinetics, and compare them to events from the same cells in the absence of the drug. Later on, the authors say that fast events are "independent of chemical transmission", but they do not show any evidence for this. All of this bears on a key question: what, if anything, do EPSPs contribute to the subthreshold events and their properties?

2) Were all of the data in Figure 1 recorded in the absence of CNQX? If the answer is yes, and CNQX reduced the frequency of those subthreshold events by 97% (see previous comment), then shouldn't we expect a large fraction of the subthreshold events to be spontaneous EPSPs rather than spikelets?

3) The recording and analyses of subthreshold events are limited by detection thresholds and signal-to-noise ratios. Some of the events (spikelets and perhaps IREs) are presumably too small to see. This has implications for the inferences one can draw from the recordings, including the estimates of coupled-neuron cluster size. Noise levels are not reported here. The authors' modeling could provide estimates of the limits of their detection given signal-to-noise limitations, and perhaps support the reliability of their cluster estimates. The authors should discuss this issue.

4) As the authors say (twice), the origins of the IREs "are a mystery". The evidence that the small spikelets are the postsynaptic result of a presynaptic action potential passing through electrical synapses is highly compelling. The smoking gun comes from paired-cell recordings that were demonstrated by various labs years ago. However, the evidence that the larger, faster "IREs" originate in axonal spines adjacent to the soma is weak, inferential, and indirect.

The modeling results about IREs are suggestive, but I agree with the authors that "The locations and mechanism of generation of the IREs are not completely resolved". In fact, without some additional experimental evidence, I do not believe this part of the story is compelling. It certainly does not live up to the overstatement in the Abstract: "fast events represent a regenerative response in unique excitable spine-like structures in the axon hillock". First, it is not even clear whether axonal spines exist in mouse IO cells. If so, they should be illustrated. The authors refer to original data on axonal spines from De Zeeuw, but that work was done on cat neurons. Second, some more direct evidence for the location of IRE generation would help to make the conclusion convincing, e.g. direct recordings from axon initial segments or spines using electrodes or photoindicators, or molecular/structural demonstrations of high-density sodium channels at axonal spines.

5) The analysis of groups of "common spikelets" in paired recordings (Figure 7) as a way to infer the size of electrically coupled clusters of cells is interesting and clever. I wonder, however, about the limitations of the data and what they mean for the cluster size estimates. One example pair is illustrated, from which four groups of common spikelets were claimed (Figure 7E). The spikelet amplitudes are small and variable, of course, and the samples of common (and noncommon) spikelets in each cell were also small (as few as two in the example pair). The Materials and methods are vague about the statistical reliability of this form of cluster counting. For example, the two spikelet clusters on the right of Figure 7E don't look very different to me, especially since the samples are so small and the spikelet sizes/shapes are so variable. The spikelet size distributions in Figure 1 also imply large and continuous variance (rather than the peaky distributions shown for the IREs). Loose criteria for spikelet clustering would lead to overestimates of the size of electrically coupled clusters. The illustrated cell pair (Figure 7E) has some of the smallest numbers of estimated connections (Figure 7F). It seems the problem of defining distinct common spikelet clusters (and estimates of connections) must also increase as the number of clusters increases in a cell. This issue requires more rigorous justification and discussion.

The modeling results in Figure 8 test the analysis in Figure 7, to some extent, by varying the sampling period and the firing frequencies. However the modeled data look much less variable (in amplitude) than the real data (cf. Figure 8B with 7B and E, for example). I also infer from the descriptions that the model network did not include noise.

6) The modeling in Figure 8 does not provide convincing support for the authors' hypothesized clustering of gap junction connectivity, as compared to a simple distance-dependent model of connectivity. This seems to boil down to another eyeball comparison of estimates from the spikelet analyses of Figure 7 to the results of a model (Figure 8). Specifically, the authors conclude that the distance model "failed to reproduce the distribution of common groups as found experimentally (Figure 8F)", whereas the clustering model "replicated the distance-dependent connection probability (Figure 8H)". To my eye the fits in Figure 8F and H seem about equally good (or bad). Interestingly, the authors conclude that the fits to data measurements in Figure 8E and H are about equally good, but to my eye the fit in Figure 8H looks about as good (or bad) as that in Figure 8F and H. In other words, the modeling results seem to be very weak evidence for deciding between these cluster vs. distance-dependence scenarios.

In this regard, it would seem that experimental studies of tracer-coupling among IO cells (as in Devor and Yarom, 2004; Placantonakis et al., 2006; Hoge et al., 2011) in the same slices as those analyzed with the common spikelet method would be a more reliable way to determine whether electrical coupling is random/distance-dependent or is determined by clustering.

7) The authors discuss the occurrence of "common spikelets", but don't say much about IREs and electrical coupling. What happens in an electrically coupled cell when an IRE occurs in the paired cell? Are these events even detectable? The Discussion simply says, "an IRE in one neuron never coincided with a spikelet in the other neuron", but I don't believe the paper tells us what does coincide with an IRE.

[Editors' note: further revisions were suggested prior to acceptance, as described below.]

Thank you for sending your article entitled "Using subthreshold events to characterize the functional architecture of the electrically coupled inferior olive network" for peer review at *eLife*. Your article is being evaluated by two peer reviewers, and the evaluation is being overseen by Ronald Calabrese as the Senior and Reviewing Editor.

The authors should be guided by the expert reviews. In particular, the authors must make their network modeling more complete (e.g., add noise) and clarify the pharmacology to make the paper strong. The clustering result and their computational method for evaluating it is potentially (pending revisions) very interesting.

Making a story about spiking axonal spines convincing appears beyond the reach of this paper, and the claims about them need to be significantly reduced or eliminated as explained in the expert reviews.

Reviewer #2:

The authors have addressed all my comments. It is a too bad that they did not provide EM pictures of the mouse axonal spines, but I find the LM sufficiently convincing.

Reviewer #3:

The authors have addressed some of the weaknesses in the original manuscript. I still think there are major shortcomings, however. The argument that IREs likely originate in axonal spines is unconvincing, in my opinion. The analysis and modeling of spikelets is a stronger story, but the modeling does not address the effects of noise in the cells. Also, the pharmacological results are still confusingly described. Specifically:

1) IRE origins and axonal spines. The authors' have slightly tempered their provocative conclusion that IREs are generated in axonal spines, although the Abstract still says: "We suggest that the fast events represent a regenerative response in unique excitable spine-like structures in the axon hillock." In my opinion the data supporting this suggestion are still exceedingly weak, for these reasons:

A) The authors' evidence for even the existence of axonal spines in mouse neurons is unconvincing. Only a single image of one mouse IO neuron was provided in the authors' response to reviews (Author response image 1). The image resolution is low, and each putative axonal spine seems to be represented by a small number of pixels. The blow-up image in Author response image 1 includes a dotted outline (hand-drawn?) that is an overly optimistic interpretation of the pixels, and the graphic in Author response image 1 is simply a cartoon version of Author response image 1. I can appreciate that imaging these small structures is technically difficult, but without knowing if spines are common, how large they are, where they are placed, and how well they correlate with the IREs, the conclusion that IREs are generated by axonal spines is simply not convincing.

B) The authors did not provide any more morphological, molecular, or electrophysiological evidence that helps to connect the origin of IREs to axonal spines (as they replied: "…we are working on this issue but currently we cannot provide this information. We hope that the demonstration of their existence is sufficient for the current report.") But their manuscript goes far beyond simply demonstrating the existence or IREs. Speculations about exotic mechanisms should wait for supporting evidence.

C) The authors simulated several possible mechanisms of IRE generation. They first showed that IRE-like events can be generated by modeling "hot spots" of excitability in the dendrites, or alternatively by simulating spike failures at increasingly distal axonal nodes. Then they dismissed these possibilities by saying "it is difficult to envisage a biological mechanism that either specifically localizes channels in a restricted dendritic 'hot spot' or that simultaneously blocks two, three or more Nodes of Ranvier" (fourth paragraph, subsection “Modelling the Intrinsic Regenerative Events (IREs)”). Perhaps so, although two papers the authors cite show evidence for relatively high densities of sodium channels in dendritic spines (Araya et al., 2007; Bywalez et al., 2015). I find it just as difficult to envisage hot spots of ion channels in biologically unique axonal spines that have not been clearly demonstrated either in the olivary cells under study or, indeed, in any other class of vertebrate neuron.

D) The modeling of spike generation in putative axonal spines (Figure 5) explored a very limited and biologically unjustified parameter space. The authors included high densities of sodium and potassium channels (identical to those in the axon initial segment) in both the spine heads and in the axon hillock, while the excitability of the axonal nodes of Ranvier was actually eliminated (the excitable hillock and the inexcitable axon are mentioned only in the Materials and methods, and not in the main text or legend). The consensus in the field is that channel densities in the axon hillocks of vertebrate neurons are quite low, especially compared to the initial segment. Perhaps mouse olivary cells are not like other neurons, but in the absence of evidence we just don't know. What was the rationale for making the rest of the axon entirely inexcitable while modeling excitable spines? Only a few results of this modeling are illustrated in the manuscript; how robust are these results? What are the consequences of varying channel densities and types, distributions, spatial patterns, spine morphology and number, etc.?

2) Network modeling and the absence of noise. The most novel and interesting conclusion from the network modeling is that the IO cells may be organized into electrically coupled clusters of cells (the connection probability predictions largely agree with the widely variable range suggested by previous studies). The clustering conclusion rests entirely on fits of the "common spikelet" distributions in recorded cell pairs to predictions of the network model. The authors note that the accuracy of the modeled connection distributions depend strongly on the cells' firing frequencies and the length of the recording samples. Should it not also depend on noise? An important feature of the biological preparation that is absent from the model is any source of noise or variability (apart from the Poisson timing of the somatic currents triggering spikes), especially in the subthreshold membrane voltages. The authors' recordings implied that blocking fast glutamate receptors reduced spontaneous spiking rates dramatically, so one can infer that there is normally a considerable of chemical synaptic noise in addition to other potential sources. How does the absence of noise in the model affect the network predictions and the fits to spikelet data, and in particular the prediction that cells are clustered?

3) Pharmacology. The pharmacology (synaptic blocker) data are still confusingly described and not very helpful. From the Results, subsection “Spontaneous unitary events recorded in neurons of the inferior olive”: "However, application of synaptic blockers (see Materials and methods) completely eliminated the presence of the fast events (n=19 neurons; in 4 of these neurons where CNQX was added during recording the frequency changed from 0.017 ± 0.005 Hz to 0 Hz) whereas the frequency of spontaneous slow events decreased significantly (from 0.92 ± 0.73 Hz 163 to 0.26 ± 0.16 Hz, p=0.019, paired t-test, n=9 neurons)." My questions:

A) What were the blockers? This Results sentence says just CNQX, the legend to Figure 2—figure supplement 1 says APV plus either CNQX or DNQX, and the Materials and methods simply list all the drugs.

B) The sample sizes tested are still ambiguous. The phrase about fast events says n=19, but then talks about n=4 "where CNQX was added". Was the drug not added to the other 15 cells? The phrase about slow events then cites n=9. Were the same cells tested before and after addition of blockers? Are the 4 and 9 cells subsets of the 19, or different samples? Please clarify.

C) The authors say they did a "thorough analysis" of the effects of blockers on slow event waveforms, but they actually report data from only two example cells in Figure 2—figure supplement 1. These data showed rise-times and half-durations, but not amplitudes.

---

## [Author Response]

Essential revisions:The experiments and modeling are well done, the analyses are interesting, and the conclusions of the study are potentially very important. There are several concerns about the main findings – the origins of the IRE and the network (neuronal cluster) analyses that must be addressed in revision. The detailed comments of the reviewers amplify these major required revisions described below; the focus in revision should be on these major points and not on minor differences in the reviewer comments.

Thank you for your encouraging response. The reviewers did a thorough job that end with important comments that undoubtedly will lead to a significant improvement of the presentation and conclusion of this study. In the revised manuscript, we re-organized the description of the data, substantially increased the description of the statistical methods used for the pharmacological treatments and modeling approach and did our best to respond to all reviewers’ comments. However, this study is an attempt to investigate the structure of electrically coupled networks by analyzing subthreshold events that represent electrical coupling. Thus, a major issue is to identify these events. In our recordings from the inferior olive neurons we encountered electrically coupled spikelets, as well as other subthreshold events, the IREs, that had to be identified before excluding them from the analysis. We suggest that they represent regenerative responses in spines that are located at the axon initial segment. We do agree that this is “a significant novel finding” that require a thorough study. In fact, we started a new line of research where anatomical, physiological and imaging technique will be used to examine our suggestion on the origin of these events. Thus, currently we can provide only anecdotal observations of spines on the axon in the mouse. A through study of spike distribution in many neurons, as well as correlation between the number of spines and the number of IREs in a given neuron and the specific activation of these spines, all are absolutely needed, and cannot be provided at this stage. Nevertheless, we did our best to reply to the reviewers’ comments and we hope that it will meet your approval.

1) All reviewers agree the synaptic pharmacology needs to be better both for the IRE and spikelet stories. The authors state they tested 19 cells in CNQX/APV, so perhaps all that's left is to do more detailed analysis and document it in the paper.

In the revised manuscript, we provided a better and more comprehensive descriptionof the pharmacological experiments. We first compared the rate of occurrence of the spontaneous events demonstrating that although the rate of spikelets are drastically reduced (n= 9 cells), the IREs are completely absent (n=4 cells). As suggested, we added a thorough description of the synaptic blockers effect on spikelets’ shape, and on appearance of evoked spikelets (Figure 2—figure supplement 1; Figure 6—figure supplement 1) to show that the reduction in frequency is due to a reduction in excitability of the network, and not because some of the spikelets we record are of synaptic origin.

The IREs, which are intrinsic events, are not encountered in the presence of synaptic blockers, but are also not of synaptic origin, as presented in Figure 2 (Figure 3 in the original manuscript). Since most of the action potentials in IO neurons are triggered by IREs, in the presence of synaptic blockers, the frequency of action potentials is highly reduced, and thus the frequency of spikelets. However, the spikelets can still be evoked with synaptic blockers, when action potentials are evoked in a coupled cell (Figure 6—figure supplement 1).

We changed the order of appearance of Figures 2 and 3 of the original manuscript in order to explain and discuss these findings.

2) Are axonal spines really as common and beautiful in mouse cells as De Zeeuw showed them to be in the cat? The authors have already dye-filled mouse IO cells, but didn't show them in the paper. These spines should be documented in the paper (LM level) and perhaps these data are already available.

Author response image 1 is a figure showing axonal spines on a mouse olivary neuron using sparse viral-labeling procedure (A). The region where the axon branches off the soma is shown in high magnification (B), and schematic illustration of the axonal spines (C). This figure can be added as a supplementary figure if needed. At this stage, we cannot provide a more detailed description of the axonal spines, as identifying the axon in IO cells is not a simple task, due to extensive bifurcations of the dendrites and the tendency of axons to emerge from a dendrite.

3) Can the authors provide more direct evidence that the IREs are coming from spines? Modeling is not enough, especially when we don't even know if axonal spines exist. Most people believe spikes initiate at axon initial segments because of high Na channel density there, and there are nice markers for initial segment-specific protein complexes (e.g. immuno for ankyrin-G, or Na channels themselves). Such a demonstration in axonal spines (LM level) by itself would be enough. Better electrophysiology (glutamate uncaging, axonal recording) would also be great, but probably too time-consuming for revision.

As we stated above, we are working on this issue but currently we cannot provide this information. We hope that the demonstration of their existence is sufficient for the current report.

4) Regarding the functional significance of the IREs, the authors never mention whether those big subthreshold IREs trigger electrotonically propagated events (small spikelets?) in coupled cells. These are data the authors should already have from their paired recordings. In general, it's not at all clear how IREs, spikelets, spontaneous oscillations, and full spikes interact in coupled networks of IO neurons; the authors should present a more holistic view of how all these membrane events interact.

In the manuscript, we specifically reported that in pair recording we *never*encountered a correlated signal with the IRE. In fact, this was our main drive to search for an alternative possibility. If the IREs were originating at dendritic level, they should have resulted in a much higher signal in the post junctional neuron. We added a clearer description of this issue in the Results section.

We added our holistic view on the interaction between these events in the Discussion.

5) The other major and potentially interesting but problematic part of the paper is about spikelets, coupling, and clustering. The authors might be able to strengthen this without doing more experiments, if they can address some of the technical issues raised in the reviews (about clarity of description, noise and detection thresholds, alternative connection schemes in the model, statistical reliability of modeling curve fits, etc).

We have addressed all technical issues, as suggested by the reviewers, listed in each specific point below.

Reviewer #1:There are several concerns:– The paper has two sections, the analysis of the subthreshold responses and the use of spikelets to analyze network connectivity. While the first section reads very well I had problems with the transition to network analysis. I found the two sentences at the start of paragraph two of subsection “Estimating network architecture from dual cell recordings of simultaneously occurring spikelets” very confusing. I had to read them several times. There is little information on how common spikelet amplitude and ratio is used to generate the groups. Although I understand the notion, the details still unclear to me. The criteria and usefulness of these values for grouping spikelets should be explicit and not let the reader guessing.

We rephrased these sentences in the Results section and in the Materials and methods sections.

– Coupling between inferior olivary neurons is heterogenous and one cell can be differentially coupled to its various neighbors. I am not sure how this would impact the analysis performed by the authors, in particular the ratio between common spikelets. Was coupling heterogeneity incorporated into computer simulations? The authors should address this issue.

We are not sure if the comment refers to heterogeneous strength of the coupling or to the heterogeneous number of neurons that are connected to each neuron. In the model, the number of neurons connected were heterogeneous (Figure 8C). The conductance of each gap-junction was identical, but the position on the dendritic tree and the cells’ morphology and membrane properties were different, thus creating heterogeneity in spikelets, as shown in Figure 8B and in figure in response to reviewer #3 comment 5.

– I suggest making the title specific to the inferior olive, perhaps ‘Using subthreshold events to characterize the functional architecture of the inferior olive’. While the number of coupled cells was not a surprise and consistent with previous reports using different methodology, the analysis performed by the authors provides further insights on the organization of the networks (favors the existence of compartments of functionally connected neurons) and this should be reflected in the title. In addition, although theoretically possible, it is unclear if the method could be used to analyze other, less known, networks. Are the authors suggesting the method could be applied other networks? If so, they should specifically discuss this point in the Discussion section.

We thank the reviewer for the title suggestion. We changed the title to be more specific to the inferior olive: 'Using subthreshold events to characterize the functional architecture of the electrically coupled inferior olive network’.

In addition, the relevance of our approach to other networks was added to the last section of Discussion section.

Reviewer #2:I have several suggestions to improve the manuscript:– The spike-rates of the IO cells were approximated, and thus a possible source of bias. Though the authors do say they estimated the firing rates based on spontaneous slow event-rates, and not on the spiking rate of the recorded cells as these are affected by the intracellular solution, I wonder to what extent the estimates of the variability and frequency of the cells' firing rates result from the cell-attached recordings.

The comment is not clear to us. To avoid artefacts of spiking activity due to the recording procedure, we estimated the firing rate from the frequency of the common spikelets that reflect the activity in single cells undisturbed by the recording. We did not use any cell-attached recordings in our experiments.

– With regard to the source of the fast spontaneous events, or "Internally Regenerative Events (IREs)", the authors find that these IREs do not depend on gap-junctions, but originate from spines in the axon hillock of the cell. I find this hypothesis quite plausible and attractive, but I wonder whether some additional electrophysiological experiments can be provided, either with glutamate uncaging or maybe glutamate or AMPA application with a secondary "puffing" pipette. This might confirm the effects of these "overruling" axonal spines on the IO neurons. More specifically, it would be nice to see an experiment with direct and/or solitary stimulation of the axonal spines.

Please see reply to editor point number 3.

– The authors suggest that the amplitudes of the IREs are predominantly determined by the neck length. However, I wonder why the observed amplitudes are not normally distributed but distributed in a sinusoid shape (see Figure 1B). I would assume that the spine neck lengths would be distributed normally. If the assumption is that the higher amplitudes are the result of summation, then I wonder why the sinusoid is so clean (because variable neck length would reduce the precision of the sinusoid shape, as the individual events that make up the summated event would be more variable as well). Additionally, as is mentioned in the text, De Zeeuw et al., 1990 showed that up to 8 spines emerge from the axon initial segment. Would the authors not expect a larger spread of amplitudes if that were the case?

Indeed, the quantal-like distribution is an interesting observation, however, this was not observed in all cells. Author response image 2) represents a different example where the amplitudes are not equally distributed.

**Author response image 2. respfig2:** 

We do not assume that the higher amplitudes are a result of summation, as this entails that both spines are innervated by the same axon. We did observe very few IREs where a break in rise-time was clearly observed (arrow in B). We assume these are the result of summation of activity in two spines. However, these IREs were omitted from analysis.

We do not expect a larger spread of amplitudes, since according to De Zeeuw et al., 1990, up to 8 spines are detected, some of them are only receiving GABAergic innervation, which will not result in IREs.

– Given that the slow spikelets predominantly occur on top of the STOs, given that action potentials in one cell often lead to spikelets in electronically coupled cells, and given that IREs often induce action potentials, wouldn't the authors expect more spikelets outside of the normal STO-spikelet-window? Or do the authors assume that the afferents to the axonal spines are somehow also influenced by the STOs? Or do the effects of the IREs depend on the phase of the subthreshold oscillations? If that were the case, would the functionality of an emergent action-potential inducer that foregoes dendritic input not be somewhat hampered?

As demonstrated in Figure 2H (former Figure 3H), as opposed to spikelets, the frequency of IREs depend on membrane potential, thus an IRE is less likely to be evoked at the trough of the STO. Moreover, it is less likely that an IRE will evoke an action potential at the trough of oscillations. Thus, although it seems that “the afferents to the axonal spines are somehow also influenced by the STOs”, it is likely to be the results of the voltage sensitivity of these events.

– There is a dependency between the underlying oscillations and the number of slow spikelets. However, spikelets would be harder to detect in steep rising phases of the oscillation. How did the authors deal with this potential confounder?

Spikelets in oscillatory traces were detected after subtracting a low pass filtered trace from the raw trace. In this way, even small spikelets on the rising and falling edges could be easily detected. We added this discerption to the Materials and methods section “Data analysis and statistics”.

– The distribution of data about amplitudes and timing is probably not normal, so it might be better to report median and quartiles rather than mean and standard deviations throughout the article.

According to Shapiro-Wilk normality test, some of the distribution are normal or close to normal, and the median/mean ratio is in some cases >0.95 and in others >0.9. Calculating the median and quartiles gave the following results, which are not very different than the mean+std:

Fast events rise time 1.4 ± 0.4 ms (median is 1.333)

Duration 4.2 ± 1.3 ms (median 3.86)

Slow events rise time 2.5 ± 0.6 ms (median 2.4)

Duration 12.7 ± 3.9 ms (median 12.6)

As such, we prefer to keep the mean+std description as this is more readable. If still needed we are happy to replace all numbers.

– I miss a plot of the distribution of STO amplitudes, particularly in the context of the claim that gap junctions are necessary for spontaneous oscillations. Our current understanding of the cell mechanisms generating the oscillations suggests that STO's may be generated in a few high amplitude cells, with other cells either reporting that oscillation, or indeed, engaging in them. Possibly in a network in which cells are connected by gap junctions a combination of these two phenomena exists. In fact Leznik and Llinas, 2004, showed that STO's are maintained within single cells following gap junction blocking, though the network synchrony is disrupted. So, in my view, it is a combination in which the level of coupling will contribute to the level of STO's, while the coupling is not absolutely essential.– The manuscript gives the impression that cells are either oscillating or not, while the literature has reported intermediate cells 'conditional oscillators'. This is an important point, since dynamical system's analysis shows that a non-oscillating cell can promptly be led to oscillate upon perturbations (Schweighofer et al., 1999).– While it is likely that oscillations are potentiated by coupling, 'network oscillations' of a couple of mV's could simply be a read-out of high amplitude oscillators. Please discuss this point more extensively.

We are puzzled by these comments. Indeed, the mechanism of generation of STO in the olivary nucleus is debated, but it is not the issue of this study. In the analysis demonstrated in Figure 6 (former Figure 2), where only stable oscillating cells were used, we used the spikelets occurrence as an indication that in this electrically coupled network all the coupled neurons oscillate at a similar frequency, which is in agreement with the different views on the source of IO oscillations, and is not discussed further.

– It would have been nice to see simulation runs with two alternate connectivity scenarios. One with fewer connections, the other with more – the latter reflecting the possible scenario in the intact preparation.

We are not sure if we completely understand the comment. In the response to comment 6 of reviewer #3 we present more connectivity scenarios.

Reviewer #3:The experiments and modeling were well done, the analyses are interesting, and the conclusions of the study are potentially important. If small spike-like events actually originate from spine-like structures on the axon initial segment, it would be a novel and unique observation as far as I know. If electrical synapse-generated spikelets can be used to reliably estimate the size and patterns of gap junction-coupled neuron clusters, that would be an important contribution to IO neurobiology and perhaps more widely. As presented here, however, I don't think either conclusion is strongly supported by the evidence.1) This study rests entirely on the properties of small, subthreshold events, so clear reporting of their characteristics is critical. A first-order question is whether any of these events represent chemical synaptic input. Are they EPSPs? Are they triggered by EPSPs? The answers here are vague. The authors report that the application of CNQX reduced the rate of "unitary unipolar events" from an average of 0.7 Hz to "below 0.02 Hz" (Results paragraph one), but no details are provided. This control rate was apparently taken from a large group of cells, not from the 19 cells to which the drug was applied. The authors should report the data properly, with measurements from the same neurons before and after CNQX. They should also report the characteristics of the subthreshold events remaining after CNQX, including amplitudes and kinetics, and compare them to events from the same cells in the absence of the drug. Later on, the authors say that fast events are "independent of chemical transmission", but they do not show any evidence for this. All of this bears on a key question: what, if anything, do EPSPs contribute to the subthreshold events and their properties?

Please see elaborated response to the editor comment 1.

2) Were all of the data in Figure 1 recorded in the absence of CNQX? If the answer is yes, and CNQX reduced the frequency of those subthreshold events by 97% (see previous comment), then shouldn't we expect a large fraction of the subthreshold events to be spontaneous EPSPs rather than spikelets?

Yes, CNQX was not used in any of the neurons presented in Figure 1. Reduction in spikelets under CNQX is explained as reduction in firing in the network. We added a new supplementary figure (Figure 2—figure supplement 1) and discussed the issue further in the Result section.

3) The recording and analyses of subthreshold events are limited by detection thresholds and signal-to-noise ratios. Some of the events (spikelets and perhaps IREs) are presumably too small to see. This has implications for the inferences one can draw from the recordings, including the estimates of coupled-neuron cluster size. Noise levels are not reported here. The authors' modeling could provide estimates of the limits of their detection given signal-to-noise limitations, and perhaps support the reliability of their cluster estimates. The authors should discuss this issue.

We agree that the analysis is restricted by the signal to noise ratio. Indeed, we discussed that our results are likely to be an underestimation of the number of spikelets and common groups. However, this limitation could only significantly affect our conclusion that the connectivity is organized in clusters. In order to test whether this will hinder our ability to reject the distance dependent configuration we reduced by half (6 pairs) the number of pairs without common groups and distributed those 6 pairs in common groups uniformly (see Author response image 3). This modified data still rejects the distance-dependent model (A, p value<0.002) and support the cluster organization (B, p value>0.05). In addition, it should be noted that in the case of 12 pairs that did not have common groups, we carefully verified that spikelets in one cell never coincide with any voltage change in the other cell.

**Author response image 3. respfig3:** 

4) As the authors say (twice), the origins of the IREs "are a mystery". The evidence that the small spikelets are the postsynaptic result of a presynaptic action potential passing through electrical synapses is highly compelling. The smoking gun comes from paired-cell recordings that were demonstrated by various labs years ago. However, the evidence that the larger, faster "IREs" originate in axonal spines adjacent to the soma is weak, inferential, and indirect.The modeling results about IREs are suggestive, but I agree with the authors that "The locations and mechanism of generation of the IREs are not completely resolved". In fact, without some additional experimental evidence, I do not believe this part of the story is compelling. It certainly does not live up to the overstatement in the Abstract: "fast events represent a regenerative response in unique excitable spine-like structures in the axon hillock". First, it is not even clear whether axonal spines exist in mouse IO cells. If so, they should be illustrated. The authors refer to original data on axonal spines from De Zeeuw, but that work was done on cat neurons. Second, some more direct evidence for the location of IRE generation would help to make the conclusion convincing, e.g. direct recordings from axon initial segments or spines using electrodes or photoindicators, or molecular/structural demonstrations of high-density sodium channels at axonal spines.

We completely agree that the source and mechanism of IRE is far from being convincing, and will need a thorough work which we are now pursuing. However, in order to decide whether to include or exclude the IREs from the organization analysis of the electrically coupled network, we had to give enough indications that they do not represent electrical coupling potentials. We added a preliminary observation on the existence of axonal spines in mouse IO cell, and amended the phrasing in the Abstract. See our elaborated response to editor comment #2.

5) The analysis of groups of "common spikelets" in paired recordings (Figure 7) as a way to infer the size of electrically coupled clusters of cells is interesting and clever. I wonder, however, about the limitations of the data and what they mean for the cluster size estimates. One example pair is illustrated, from which four groups of common spikelets were claimed (Figure 7E). The spikelet amplitudes are small and variable, of course, and the samples of common (and noncommon) spikelets in each cell were also small (as few as two in the example pair). The Materials and methods are vague about the statistical reliability of this form of cluster counting. For example, the two spikelet clusters on the right of Figure 7E don't look very different to me, especially since the samples are so small and the spikelet sizes/shapes are so variable. The spikelet size distributions in Figure 1 also imply large and continuous variance (rather than the peaky distributions shown for the IREs). Loose criteria for spikelet clustering would lead to overestimates of the size of electrically coupled clusters. The illustrated cell pair (Figure 7E) has some of the smallest numbers of estimated connections (Figure 7F). It seems the problem of defining distinct common spikelet clusters (and estimates of connections) must also increase as the number of clusters increases in a cell. This issue requires more rigorous justification and discussion.

We agree with the reviewer that incorrect spikelet clustering will hamper our estimation of connected cells, and for this reason we took every effort to devise an analysis that will correctly cluster the traces. It is true that, as shown in Figure 1, the distribution of spikelets’ amplitudes is large. However, when taking the 3 parameters: amplitude of common spikelet in the first cell; amplitude of common spikelet in the second cell; difference between the two amplitudes, one gets a very clear spikelet clusters.

In both the experimental and modelling parts we initially used the DBSCAN clustering algorithm to cluster the common spikelets according to their peak amplitudes and differance between amplitudes. However, while the algorithm works well for the model, it is less ideal for the noisy experimental traces although the number of groups detected by the algorithm was not substantially different than the manual curation. In Author response image 4 we compare the manual curation to the DBSCAN results in another example pair. Whereas the similarity of two modes of analysis are clearly seen, the algorithm predicted 7 groups whereas the manual analysis resulted in 6 groups. First, it is clear that the manual curation is more accurate. Second, as we demonstrate above (response to comment 3), a small change in the number of common groups did not affect the significance of our claim.

In the example shown in Figure 7, the two right clusters indeed have similar black traces, but the red traces could easily be grouped in two different clusters according to their amplitudes. We chose this example, since we believe it is an easy example to explain our methodological approach with.

We added a clearer description of the analysis in the Materials and methods section.

**Author response image 4. respfig4:** 

The modeling results in Figure 8 test the analysis in Figure 7, to some extent, by varying the sampling period and the firing frequencies. However the modeled data look much less variable (in amplitude) than the real data (cf. Figure 8B with 7B and E, for example). I also infer from the descriptions that the model network did not include noise.

Indeed, in the example given in Figure 8 there is little variance, however, this was not a common feature, as shown in another example in Author response image 5. The difference in spikelets’ amplitude was originated from the difference in GJ locations and the cells’ morphology and membrane properties. There was also noise originating from the difference in the mean firing rate of the cells, which effect we quantified in Figure 8—figure supplement 1. Noise was not added to the voltage traces.

**Author response image 5. respfig5:** 

6) The modeling in Figure 8 does not provide convincing support for the authors' hypothesized clustering of gap junction connectivity, as compared to a simple distance-dependent model of connectivity. This seems to boil down to another eyeball comparison of estimates from the spikelet analyses of Figure 7 to the results of a model (Figure 8). Specifically, the authors conclude that the distance model "failed to reproduce the distribution of common groups as found experimentally (Figure 8F)", whereas the clustering model "replicated the distance-dependent connection probability (Figure 8H)". To my eye the fits in Figure 8F and H seem about equally good (or bad). Interestingly, the authors conclude that the fits to data measurements in Figure 8E and H are about equally good, but to my eye the fit in Figure 8H looks about as good (or bad) as that in Figure 8F and H. In other words, the modeling results seem to be very weak evidence for deciding between these cluster vs. distance-dependence scenarios.In this regard, it would seem that experimental studies of tracer-coupling among IO cells (as in Devor and Yarom, 2004; Placantonakis et al., 2006; Hoge et al., 2011) in the same slices as those analyzed with the common spikelet method would be a more reliable way to determine whether electrical coupling is random/distance-dependent or is determined by clustering.

Thank you for the comment. We conducted a full statistical analysis on this issue which is discussed in a new section in the Materials and methods and a supplementary figure (Figure 8 —figure supplement 2).

In short, we added a p-value for Figure 8F and 8I (<0.002 for 8F, and 0.907 and 0.125 for 8I black and gray, respectively). And also for Figure 8E and H (all > 0.05).

Furthermore, in the original manuscript we tested the common group distribution only for one distance dependent connectivity configuration. Here we validated that our results are not constrained only to that specific distant-dependent configuration, as also pointed by the two other reviewers. In Author response image 6 are connectivity analysis of networks with different distance dependent connectivity probability (left), in which the Σ and σ (see Materials and methods, connectivity matrices) are different, and the corresponding common neighbor distribution (right). For all those connectivity configurations, the p values for the common neighbor distribution were <0.05.

**Author response image 6. respfig6:** 

7) The authors discuss the occurrence of "common spikelets", but don't say much about IREs and electrical coupling. What happens in an electrically coupled cell when an IRE occurs in the paired cell? Are these events even detectable? The Discussion simply says, "an IRE in one neuron never coincided with a spikelet in the other neuron", but I don't believe the paper tells us what does coincide with an IRE.

We did not see any event or change in membrane potential in one cell while IRE was apparent in the other cell, nor we saw an IRE in one cell due to spike in the other cell. These evidences highly support the hypothesis that IREs are not evoked in the dendrites, as this will pass through gap-junction to coupled neurons. This was indeed our reason to search for other possibility to the origin of the IREs, after dismissing the direct-indirect spikelets possibility which is presented in Figure 4—figure supplement 1. This point was elaborated in the Result section and Discussion.

[Editors' note: further revisions were suggested prior to acceptance, as described below.]

The major change of the current submission is a complete removal of the axonal spines and focus on the organization of the network.

In the current version we also answered the reviewer’s comments by:

a) Performed a new set of pharmacological experiments, showing that the fast events are blocked by synaptic blockers whereas the spikelets are still present and therefore are likely to reflect electrical coupling. As mentioned by the reviewers, the fact that the frequency of spikelets is significantly reduced by the drugs may suggest that some of the slow subthreshold events might be a chemical synaptic event, a possibility that will question the use of spontaneous events to study the network organization. To resolve this issue, we first compared the spontaneous spikelets to the evoked ones (in pair recordings) and show their similarity. Second, we analyzed a new set of experiments of light-evoked chemical synapses in the Thy1 mouse, demonstrating that their waveform and amplitude are different from the spontaneous and evoked slow events. Thus, we conclude that most, if not all, of the spontaneous slow events are indeed due to electrical coupling.

In addition, we discuss the reduction in spikelets activity under drug conditions, arguing that spontaneous spiking activity is triggered by the fast events and that blocking the fast events result in a reduction in the frequency of spikelets.

b) We study the role of noise in detecting common groups of spikelets by assuming either failure or over detections. This is summarized in Figure 7. We also added simulation of the expected distribution of common groups under different assumptions. In all these simulations it is rather clear that a network where the probability of connections is only distance dependent cannot account for the experimentally observed distribution. Thus the main conclusion of this study, that the inferior olive nucleus is organized in clustered of connected cells, is fully supported by the experimental observations.

In our opinion, it is convincing that we offer a novel experimental and theoretical approach to the study of electrically-coupled network and hope it will meet your approval.

Reviewer #3:The authors have addressed some of the weaknesses in the original manuscript. I still think there are major shortcomings, however. The argument that IREs likely originate in axonal spines is unconvincing, in my opinion. The analysis and modeling of spikelets is a stronger story, but the modeling does not address the effects of noise in the cells. Also, the pharmacological results are still confusingly described.1) IRE origins and axonal spines. The authors' have slightly tempered their provocative conclusion that IREs are generated in axonal spines, although the Abstract still says: "We suggest that the fast events represent a regenerative response in unique excitable spine-like structures in the axon hillock." In my opinion the data supporting this suggestion are still exceedingly weak, for these reasons:A) The authors' evidence for even the existence of axonal spines in mouse neurons is unconvincing. Only a single image of one mouse IO neuron was provided in the authors' response to reviews (Author response image 1). The image resolution is low, and each putative axonal spine seems to be represented by a small number of pixels. The blow-up image in Author response image 1 includes a dotted outline (hand-drawn?) that is an overly optimistic interpretation of the pixels, and the graphic in Author response image 1 is simply a cartoon version of Author response image 1. I can appreciate that imaging these small structures is technically difficult, but without knowing if spines are common, how large they are, where they are placed, and how well they correlate with the IREs, the conclusion that IREs are generated by axonal spines is simply not convincing.B) The authors did not provide any more morphological, molecular, or electrophysiological evidence that helps to connect the origin of IREs to axonal spines (as they replied: "…we are working on this issue but currently we cannot provide this information. We hope that the demonstration of their existence is sufficient for the current report.") But their manuscript goes far beyond simply demonstrating the existence or IREs. Speculations about exotic mechanisms should wait for supporting evidence.C) The authors simulated several possible mechanisms of IRE generation. They first showed that IRE-like events can be generated by modeling "hot spots" of excitability in the dendrites, or alternatively by simulating spike failures at increasingly distal axonal nodes. Then they dismissed these possibilities by saying "it is difficult to envisage a biological mechanism that either specifically localizes channels in a restricted dendritic 'hot spot' or that simultaneously blocks two, three or more Nodes of Ranvier" (fourth paragraph, subsection “Modelling the Intrinsic Regenerative Events (IREs)”). Perhaps so, although two papers the authors cite show evidence for relatively high densities of sodium channels in dendritic spines (Araya et al., 2007; Bywalez et al., 2015). I find it just as difficult to envisage hot spots of ion channels in biologically unique axonal spines that have not been clearly demonstrated either in the olivary cells under study or, indeed, in any other class of vertebrate neuron.D) The modeling of spike generation in putative axonal spines (Figure 5) explored a very limited and biologically unjustified parameter space. The authors included high densities of sodium and potassium channels (identical to those in the axon initial segment) in both the spine heads and in the axon hillock, while the excitability of the axonal nodes of Ranvier was actually eliminated (the excitable hillock and the inexcitable axon are mentioned only in the Materials and methods, and not in the main text or legend). The consensus in the field is that channel densities in the axon hillocks of vertebrate neurons are quite low, especially compared to the initial segment. Perhaps mouse olivary cells are not like other neurons, but in the absence of evidence we just don't know. What was the rationale for making the rest of the axon entirely inexcitable while modeling excitable spines? Only a few results of this modeling are illustrated in the manuscript; how robust are these results? What are the consequences of varying channel densities and types, distributions, spatial patterns, spine morphology and number, etc.?

As mentioned above this part has been removed.

2) Network modeling and the absence of noise. The most novel and interesting conclusion from the network modeling is that the IO cells may be organized into electrically coupled clusters of cells (the connection probability predictions largely agree with the widely variable range suggested by previous studies).

Thank you. We certainly agree with this description.

The clustering conclusion rests entirely on fits of the "common spikelet" distributions in recorded cell pairs to predictions of the network model. The authors note that the accuracy of the modeled connection distributions depend strongly on the cells' firing frequencies and the length of the recording samples. Should it not also depend on noise? An important feature of the biological preparation that is absent from the model is any source of noise or variability (apart from the Poisson timing of the somatic currents triggering spikes), especially in the subthreshold membrane voltages.

We study the role of noise in detecting common groups of spikelets by assuming either failure or over detections. This is summarized in the new Figure 7, showing that even if we assume a failure in detecting common group, or inaccuracy in the number of common groups detection, the clustered organization is still the only possible explanation. Furthermore, we examined several distance-dependent connectivity possibilities and none of them produce a distribution of common groups that fits the experimental results. Thus we can strongly conclude that the only explanation for the observed distribution is clustered organization.

The authors' recordings implied that blocking fast glutamate receptors reduced spontaneous spiking rates dramatically, so one can infer that there is normally a considerable of chemical synaptic noise in addition to other potential sources.

As mentioned above, the reduction in spikelets activity is highly correlated with reduction in the fast events which in the IO trigger spiking activity (Figure 3—figure supplement 2). Thus blocking the fast events will lead to a reduction in spiking activity followed by a reduction in spikelets. The synaptic potentials that trigger the fast events cannot be recorded at somatic location. Spontaneous synaptic events are rarely encountered. However, to examine the possible involvement of chemical synapses in the spontaneous activity we analyzed synaptic events that were triggered by ChR activation of excitatory axons projecting to IO neurons in Thy1 mice. Using minimal light duration we were able to activate putatively unitary events. We then compared the waveforms of the synaptic potentials to the waveforms of the slow events. As shown in Figure 3—figure supplement 1, the synaptic potentials’ waveform and amplitude are clearly different from the spontaneous slow event (n=17 cells). Furthermore, we demonstrate (Figure 2) that the spikelets are insensitive to membrane potential. This, which has been examined in large population of neurons, further supports our conclusion that most, if not all, of the slow events are a reflection of electrical coupling. One can argue that the insensitivity to membrane potential might be due to distal location of the synapse, however, the shape of the spikelets does not support such possibility.

3) Pharmacology. The pharmacology (synaptic blocker) data are still confusingly described and not very helpful. From the Results, subsection “Spontaneous unitary events recorded in neurons of the inferior olive”: "However, application of synaptic blockers (see Materials and methods) completely eliminated the presence of the fast events (n=19 neurons; in 4 of these neurons where CNQX was added during recording the frequency changed from 0.017 ± 0.005 Hz to 0 Hz) whereas the frequency of spontaneous slow events decreased significantly (from 0.92 ± 0.73 Hz 163 to 0.26 ± 0.16 Hz, p=0.019, paired t-test, n=9 neurons)." My questions:A) What were the blockers? This Results sentence says just CNQX, the legend to Figure 2—figure supplement 1 says APV plus either CNQX or DNQX, and the Materials and methods simply list all the drugs.B) The sample sizes tested are still ambiguous. The phrase about fast events says n=19, but then talks about n=4 "where CNQX was added". Was the drug not added to the other 15 cells? The phrase about slow events then cites n=9. Were the same cells tested before and after addition of blockers? Are the 4 and 9 cells subsets of the 19, or different samples? Please clarify.C) The authors say they did a "thorough analysis" of the effects of blockers on slow event waveforms, but they actually report data from only two example cells in Figure 2—figure supplement 1. These data showed rise-times and half-durations, but not amplitudes.

We added a new cohort of data and simplified the description of the drug effect. As mentioned above, we show that the drug completely eliminated the fast events accompanied by significant reduction in spikelets activity. This has been added to the text, and to Figure 3—figure supplement 1, and we hope it is now clear.